# Proteome profile of peripheral myelin in healthy mice and in a neuropathy model

Sophie B Siems[1][†], Olaf Jahn[2][†], Maria A Eichel[1], Nirmal Kannaiyan[3], Lai Man N Wu[4], Diane L Sherman[4], Kathrin Kusch[1], Dörte Hesse[2], Ramona B Jung[1], Robert Fledrich[1,5], Michael W Sereda[1,6], Moritz J Rossner[3], Peter J Brophy[4], Hauke B Werner[1]*

[1]Department of Neurogenetics, Max Planck Institute of Experimental Medicine, Göttingen, Germany; [2]Proteomics Group, Max Planck Institute of Experimental Medicine, Göttingen, Germany; [3]Department of Psychiatry and Psychotherapy, University Hospital, LMU Munich, Munich, Germany; [4]Centre for Discovery Brain Sciences, University of Edinburgh, Edinburgh, United Kingdom; [5]Institute of Anatomy, University of Leipzig, Leipzig, Germany; [6]Department of Clinical Neurophysiology, University Medical Center, Göttingen, Germany

**Abstract** Proteome and transcriptome analyses aim at comprehending the molecular profiles of the brain, its cell-types and subcellular compartments including myelin. Despite the relevance of the peripheral nervous system for normal sensory and motor capabilities, analogous approaches to peripheral nerves and peripheral myelin have fallen behind evolving technical standards. Here we assess the peripheral myelin proteome by gel-free, label-free mass-spectrometry for deep quantitative coverage. Integration with RNA-Sequencing-based developmental mRNA-abundance profiles and neuropathy disease genes illustrates the utility of this resource. Notably, the periaxin-deficient mouse model of the neuropathy Charcot-Marie-Tooth 4F displays a highly pathological myelin proteome profile, exemplified by the discovery of reduced levels of the monocarboxylate transporter MCT1/SLC16A1 as a novel facet of the neuropathology. This work provides the most comprehensive proteome resource thus far to approach development, function and pathology of peripheral myelin, and a straightforward, accurate and sensitive workflow to address myelin diversity in health and disease.

***For correspondence:**
hauke@em.mpg.de

[†]These authors contributed equally to this work

**Competing interests:** The authors declare that no competing interests exist.

## Introduction

The ensheathment of axons with myelin enables rapid impulse propagation, a prerequisite for normal motor and sensory capabilities of vertebrates (*Weil et al., 2018*; *Hartline and Colman, 2007*). This is illustrated by demyelinating neuropathies of the Charcot-Marie-Tooth (CMT) spectrum, in which mutations affecting myelin genes as *MPZ, PMP22, GJB1* and *PRX* impair myelin integrity and reduce the velocity of nerve conduction in the peripheral nervous system (PNS) (*Rossor et al., 2013*). Developmentally, myelination by Schwann cells in peripheral nerves is regulated by axonal neuregulin-1 (*Michailov et al., 2004*; *Taveggia et al., 2005*) and the basal lamina (*Chernousov et al., 2008*; *Petersen et al., 2015*; *Ghidinelli et al., 2017*) that is molecularly linked to the abaxonal Schwann cell membrane via integrins and the dystroglycan complex (*Sherman et al., 2001*; *Masaki et al., 2002*; *Nodari et al., 2008*; *Raasakka et al., 2019*). In adulthood, the basal lamina continues to enclose all axon/myelin-units (*Hess and Lansing, 1953*), probably to maintain myelin. Beyond regulation by extracellular cues, myelination involves multiple proteins mediating radial sorting of axons out of Remak bundles, myelin membrane growth and layer compaction (*Sherman and Brophy, 2005*; *Pereira et al., 2012*; *Grove and Brophy, 2014*; *Monk et al., 2015*; *Feltri et al., 2016*). For example, the Ig-domain containing myelin protein zero

(MPZ; also termed P0) mediates adhesion between adjacent extracellular membrane surfaces in compact myelin (*Giese et al., 1992*). At their intracellular surfaces, myelin membranes are compacted by the cytosolic domain of MPZ/P0 together with myelin basic protein (MBP; previously termed P1) (*Martini et al., 1995*; *Nawaz et al., 2013*). Not surprisingly, MPZ/P0 and MBP were early identified as the most abundant peripheral myelin proteins (*Greenfield et al., 1973*; *Brostoff et al., 1975*).

A system of cytoplasmic channels through the otherwise compacted myelin sheath remains non-compacted throughout life, that is the adaxonal myelin layer, paranodal loops, Schmidt-Lanterman incisures (SLI), and abaxonal longitudinal and transverse bands of cytoplasm termed bands of Cajal (*Sherman and Brophy, 2005*; *Nave and Werner, 2014*; *Kleopa and Sargiannidou, 2015*). Non-compacted myelin comprises cytoplasm, cytoskeletal elements, vesicles and lipid-modifying enzymes, and thus numerous proteins involved in maintaining the myelin sheath. The cytosolic channels probably also represent transport routes toward Schwann cell-dependent metabolic support of myelinated axons (*Court et al., 2004*; *Beirowski et al., 2014*; *Domènech-Estévez et al., 2015*; *Kim et al., 2016*; *Gonçalves et al., 2017*; *Stassart et al., 2018*).

Considering that Schwann cells constitute a major proportion of the cells in the PNS, oligonucleotide microarray analyses have been used for mRNA abundance profiling of total sciatic nerves (*Nagarajan et al., 2002*; *Le et al., 2005*). Indeed, these systematic approaches allowed the identification of novel myelin constituents including non-compact myelin-associated protein (NCMAP/MP11) (*Ryu et al., 2008*). Notwithstanding that the number of known peripheral myelin proteins has grown in recent years, a comprehensive molecular inventory has been difficult to achieve because applications of systematic ('omics') approaches specifically to Schwann cells and peripheral myelin remained comparatively scarce, different from studies addressing oligodendrocytes and CNS myelin (*Zhang et al., 2014*; *Patzig et al., 2016b*; *Sharma et al., 2015*; *Thakurela et al., 2016*; *Marques et al., 2016*; *de Monasterio-Schrader et al., 2012*). One main reason may be that the available techniques were not sufficiently straightforward for general application. For example, the protein composition of peripheral myelin was previously assessed by proteome analysis (*Patzig et al., 2011*). However, at that time the workflow of sample preparation and data acquisition (schematically depicted in *Figure 1A*) was very labor-intense and required a substantial amount of input material; yet the depth of the resulting datasets remained limited. In particular, differential myelin proteome analysis by 2-dimensional fluorescence intensity gel electrophoresis (2D-DIGE) requires considerable hands-on-time and technical expertise (*Patzig et al., 2011*; *Kangas et al., 2016*). While this method is powerful for the separation of proteoforms (*Kusch et al., 2017*), it typically suffers from under-representation of highly basic and transmembrane proteins. It thus allows comparing the abundance of only few myelin proteins rather than quantitatively covering the entire myelin proteome. Because of these limitations and an only modest sample-to-sample reproducibility, 2D-DIGE analysis of myelin, although unbiased, has not been commonly applied beyond specialized laboratories.

The aim of the present study was to establish a straightforward and readily applicable workflow to facilitate both comprehensive knowledge about the protein composition of peripheral myelin and systematic assessment of differences between two states, for example, pathological alterations in a neuropathy model. The major prerequisites were the biochemical purification of myelin, its solubilization with the detergent ASB-14 and the subsequent automated digestion with trypsin during filter-aided sample preparation (FASP). The tryptic peptides were fractionated by liquid chromatography and analyzed by mass spectrometry for gel-free, label-free quantitative proteome analysis. More specifically, we used nano-flow ultra-performance liquid chromatography (nanoUPLC) coupled to an electrospray-ionization quadrupole time-of-flight (ESI-QTOF) mass spectrometer with ion mobility option, providing an orthogonal dimension of peptide separation. The utilized data-independent acquisition (DIA) strategy relies on collecting data in an alternating low and elevated energy mode (MS$^E$); it enables simultaneous sequencing and quantification of all peptides entering the mass spectrometer without prior precursor selection, as reviewed in *Neilson et al. (2011)* and *Distler et al. (2014a)*. With their high-duty cycle utilized for the acquisition of precursor ions, MS$^E$-type methods are ideally suited to reliably quantify proteins based on peptide intensities. Notably, these methods do not involve the use of spectral libraries in the identification of proteins, different from other DIA strategies. Instead, the achieved high-complexity fragmentation spectra are deconvoluted before submission to dedicated search engines for peptide and protein identification (*Geromanos et al.,*

  

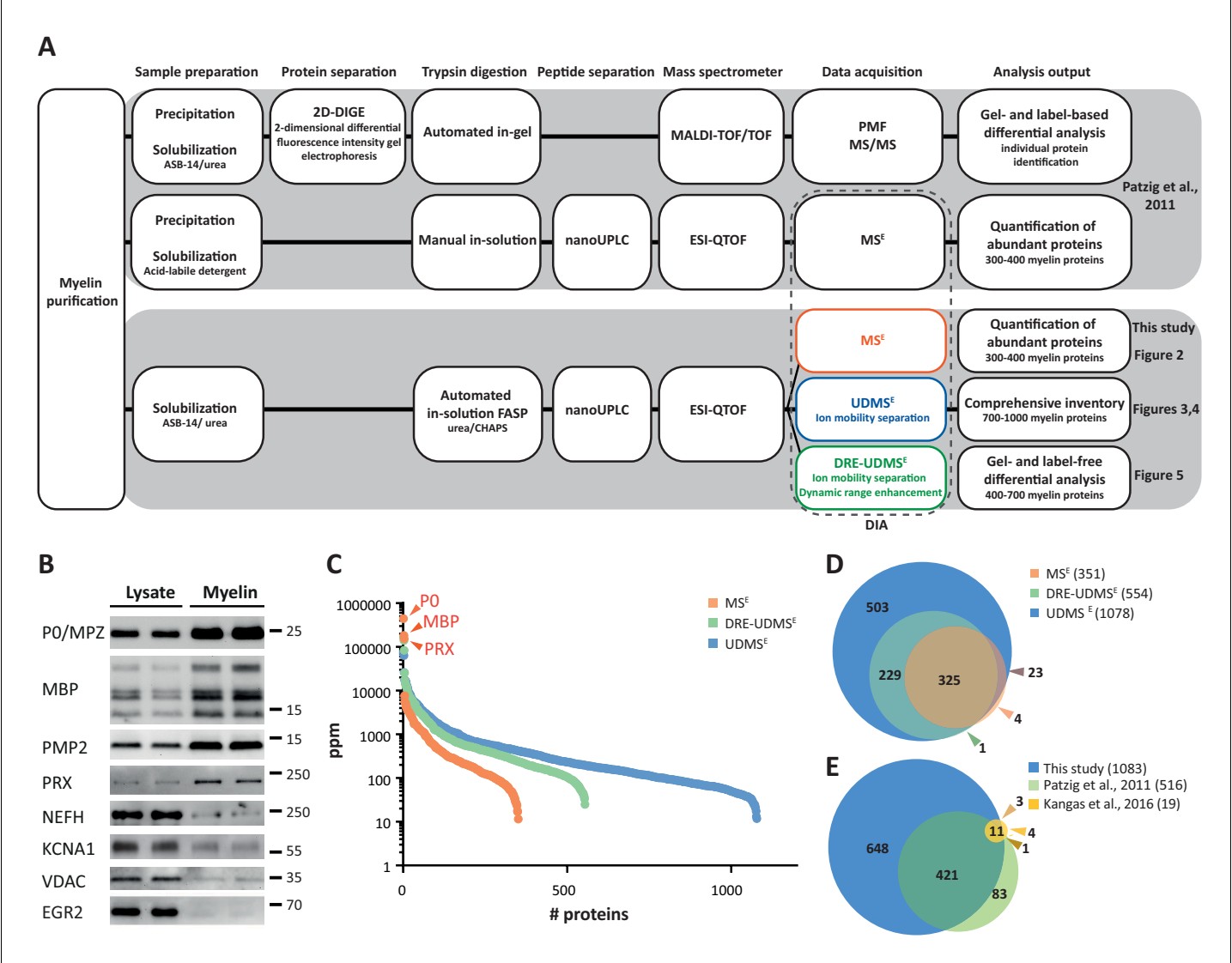

**Figure 1.** Proteome analysis of peripheral myelin. (**A**) Schematic illustration of a previous approach to the peripheral myelin proteome (*Patzig et al., 2011*) compared with the present workflow. Note that the current workflow allows largely automated sample processing and omits labor-intense 2-dimensional differential gel-electrophoresis, thereby considerably reducing hands-on time. Nano LC-MS analysis by data-independent acquisition (DIA) using three different data acquisition modes provides efficient identification and quantification of abundant myelin proteins (MS[E]; see *Figure 2*), a comprehensive inventory (UDMS[E]; see *Figures 3–4*) and gel-free differential analysis of hundreds of distinct proteins (DRE-UDMS[E]; see *Figure 5*). Samples were analyzed in three biological replicates. (**B**) Immunoblot of myelin biochemically enriched from sciatic nerves of wild-type mice at postnatal day 21 (P21). Equal amounts of corresponding nerve lysate were loaded to compare the abundance of marker proteins for compact myelin (MPZ/P0, MBP, PMP2), non-compact myelin (PRX), the Schwann cell nucleus (KROX20/EGR2), axons (NEFH, KCNA1) and mitochondria (VDAC). Blots show n = 2 biological replicates representative of n = 3 biological replicates. Note that myelin markers are enriched in purified myelin; other cellular markers are reduced. (**C**) Number and relative abundance of proteins identified in myelin purified from the sciatic nerves of wild-type mice using three different data acquisition modes (MS[E], UDMS[E], DRE-UDMS[E]). Note that MS[E] (orange) provides the best information about the relative abundance of high-abundant myelin proteins (dynamic range of more than four orders of magnitude) but identifies comparatively fewer proteins in purified myelin. UDMS[E] (blue) identifies the largest number of proteins but provides only a lower dynamic range of about three orders of magnitude. DRE-UDMS[E] (green) identifies an intermediate number of proteins with an intermediate dynamic range of about four orders of magnitude. Note that MS[E] with very high dynamic range is required for the quantification of the exceptionally abundant myelin protein zero (MPZ/P0), myelin basic protein (MBP) and periaxin (PRX). ppm, parts per million. (**D**) Venn diagram comparing the number of proteins identified in PNS myelin by MS[E], UDMS[E] and DRE-UDMS[E]. Note the high overlap of identified proteins. (**E**) Venn diagram of the proteins identified in PNS myelin by UDMS[E] in this study compared with those identified in two previous approaches (*Patzig et al., 2011*; *Kangas et al., 2016*).

The online version of this article includes the following source data and figure supplement(s) for figure 1:

*Figure 1 continued on next page*

*Figure 1 continued*

**Source data 1.** Label-free quantification of proteins in wild-type PNS myelin fractions by three different data acquisition modes Identification and quantification data of detected myelin-associated proteins.

**Figure supplement 1.** Clustered heatmap of Pearson's correlation coefficients for protein abundance comparing data acquisition modes.

*2009*; *Li et al., 2009*). In the MS$^E$ mode, this deconvolution involves precursor-fragment ion alignment solely on the basis of chromatographic elution profiles. On top, drift times of ion mobility-separated precursors are used in the high-definition (HD)MS$^E$ mode. An expansion of the latter, referred to as the ultra-definition (UD)MS$^E$ mode, additionally implements drift time-dependent collision energy profiles for more effective precursor fragmentation (*Distler et al., 2016*; *Distler et al., 2014b*).

Indeed, compared to the previously used manual handling and in-gel digestion, the current workflow (schematically depicted in *Figure 1A*) is considerably less labor-intense, and automated FASP increases sample-to-sample reproducibility. Moreover, differential analysis by quantitative mass spectrometry (MS) facilitates reproducible quantification of hundreds rather than a few distinct myelin proteins. Together, the present workflow increases the efficacy of assessing the peripheral myelin proteome while shifting the main workload from manual sample preparation and gel-separation to automated acquisition and processing of data. We propose that comprehending the expression profiles of all myelin proteins in the healthy PNS and in myelin-related disorders can contribute to advancing our understanding of the physiology and pathophysiology of peripheral nerves.

## Results

### Purification of peripheral myelin

We biochemically enriched myelin as a light-weight membrane fraction from pools of sciatic nerves dissected from mice at postnatal day 21 (P21) using an established protocol of discontinuous sucrose density gradient centrifugation (*Patzig et al., 2011*; *Larocca and Norton, 2006*), in which myelin membranes accumulate at the interface between 0.29 and 0.85 M sucrose. By immunoblotting, proteins specific for both compact (MPZ/P0, MBP, PMP2) and non-compact (PRX) myelin were substantially enriched in the myelin fraction compared to nerve lysates (*Figure 1B*). Conversely, axonal (NEFH, KCNA1) and mitochondrial (VDAC) proteins and a marker for the Schwann cell nucleus (KROX20/EGR2) were strongly reduced in purified myelin. Together, these results imply that biochemically purified peripheral myelin is suitable for systematic analysis of its protein composition.

### Proteome analysis of peripheral myelin

It has long been difficult to accurately quantify the most abundant myelin proteins both in the CNS (PLP, MBP, CNP; *Jahn et al., 2009*) and the PNS (MPZ/P0, MBP, PRX; this work), probably owing to their exceptionally high relative abundance. For example, the major CNS myelin constituents PLP, MBP and CNP comprise 17, 8 and 4% of the total myelin protein, respectively (*Jahn et al., 2009*). We have recently provided proof of principle (*Erwig et al., 2019a*) that the mass spectrometric quantification of these high-abundant myelin proteins is accurate and precise when data are acquired in the MS$^E$ data acquisition mode and proteins are quantified according to the TOP3 method, that is if their abundance values are obtained based on the proven correlation between the average intensity of the three peptides exhibiting the most intense mass spectrometry response and the absolute amount of their source protein (*Silva et al., 2006*; *Ahrné et al., 2013*). Using data acquisition by MS$^E$ we confirmed that CNP constitutes about 4% of the total CNS myelin proteome and that the abundance of CNP in myelin from mice heterozygous for the *Cnp* gene (*Cnp*$^{WT/null}$) compared to wild-type mice is 50.7% (±0.4%), in agreement with the halved gene dosage and gel-based quantification by silver staining or immunoblotting (*Erwig et al., 2019a*).

When applying the MS$^E$ mode to PNS myelin, we quantified 351 proteins with a false discovery rate (FDR) of <1% at peptide and protein level and an average sequence coverage of 35.5% (*Figure 1—source data 1*). While MS$^E$ (labeled in orange in *Figure 1C*) indeed provided a dynamic range of more than four orders of magnitude and thus quantitatively covered the exceptionally abundant myelin proteins MPZ/P0, MBP and PRX, the number of quantified proteins appeared

limited when spectral complexity was deconvoluted solely on the basis of chromatographic elution profiles. Accordingly, by using the UDMS$^E$ mode, which comprises ion mobility for additional peptide separation as well as drift time-specific collision energies for peptide fragmentation, proteome coverage was increased about three-fold (1078 proteins quantified; average sequence coverage 34.3%; *Figure 1—source data 1*). However, the dynamic range of UDMS$^E$ (labeled in blue in *Figure 1C*) was found to be somewhat compressed compared to that of MS$^E$, which can be considered an expectable feature of traveling wave ion mobility devices (*Dodds and Baker, 2019*), where the analysis of pulsed ion packages leads to a temporal and spatial binning of peptides during ion mobility separation. Indeed, this manifests as a ceiling effect for the detection of exceptionally intense peptide signals, which results in an underestimation of the relative abundance of MPZ/P0, MBP and PRX by UDMS$^E$.

The complementary nature of the MS$^E$ and UDMS$^E$ data acquisition modes led us to conclude that a comprehensive analysis of the myelin proteome that facilitates both correct quantification of the most abundant proteins and deep quantitative coverage of the proteome would require analyzing the same set of samples with two different instrument settings for MS$^E$ and UDMS$^E$, respectively. Considering that instrument time is a bottleneck for the routine differential proteome analysis of myelin from mutant mice, we aimed to combine the strengths of MS$^E$ and UDMS$^E$ into a single data acquisition mode. Based on a gene ontology enrichment analysis for cellular components of the 200 proteins of highest and lowest abundance from the UDMS$^E$ dataset, we realized that the 'bottom 'of the quantified proteome is probably largely unrelated to myelin but dominated by contaminants from other subcellular sources including mitochondria. We thus reasoned that for a myelin-directed data acquisition mode, proteome depth may be traded in for a gain in dynamic range and devised a novel method referred to as dynamic range enhancement (DRE)-UDMS$^E$, in which a deflection lens is used to cycle between full and reduced ion transmission during mass spectrometric scanning. Indeed, DRE-UDMS$^E$ quantified an intermediate number of proteins in PNS myelin (554 proteins; average sequence coverage 30.6%; *Figure 1—source data 1*) while providing an intermediate dynamic range (labeled in green in *Figure 1C*). We thus consider DRE-UDMS$^E$ as the data acquisition mode of choice most suitable for routine differential myelin proteome profiling (see below).

Overall, we found a high reproducibility between replicates and even among the different data acquisition modes as indicated by Pearson's correlation coefficients for protein abundance in the range of 0.765–0.997 (*Figure 1—figure supplement 1*). When comparing the proteins identified in PNS myelin using the three data acquisition modes, we found a very high overlap (*Figure 1D*). We also found a high overlap (*Figure 1E*) between the proteins identified in the present study by UDMS$^E$ and those detected in previous proteomic approaches to PNS myelin (*Patzig et al., 2011*; *Kangas et al., 2016*), thus allowing a high level of confidence. Together, the three data acquisition modes exhibit distinct strengths in the efficient quantification of exceptionally abundant proteins (MS$^E$), establishing a comprehensive inventory (UDMS$^E$) and gel-free, label-free differential analysis of hundreds of distinct proteins (DRE-UDMS$^E$) in peripheral myelin (see *Figure 1A*). Yet, analyzing the same set of samples by different modes may not always be feasible in all routine applications when considering required instrument time.

## Relative abundance of peripheral myelin proteins

Considering that MS$^E$ provides the high dynamic range required for the quantification of the most abundant myelin proteins, we calculated the relative abundance of the 351 proteins identified in myelin by MS$^E$ (*Figure 1—source data 1*). According to quantitative assessment of this dataset, the most abundant PNS myelin protein, myelin protein zero (MPZ/P0), constitutes 44% (+/- 4% relative standard deviation (RSD)) of the total myelin protein (*Figure 2*). Myelin basic protein (MBP), periaxin (PRX) and tetraspanin-29 (CD9) constitute 18% (+/- 1% RSD), 15% (+/- 1%) and 1% (+/- 0.2%) of the total myelin protein, respectively (*Figure 2*). For MPZ/P0 and MBP, our quantification by MS$^E$ is in agreement with but specifies prior estimations upon gel-separation and protein labeling by Sudan-Black, Fast-Green or Coomassie-Blue, in which they were judged to constitute 45–70% and 2–26% of the total myelin protein, respectively (*Greenfield et al., 1973*; *Micko and Schlaepfer, 1978*; *Smith and Curtis, 1979*; *Whitaker, 1981*). However, gel-based estimates of the relative abundance of myelin proteins were not very precise with respect to many other proteins, including those of high molecular weight. Indeed, periaxin was identified as a constituent of peripheral myelin after the advent of gradient SDS-PAGE gels (*Gillespie et al., 1994*), which allowed improved migration of

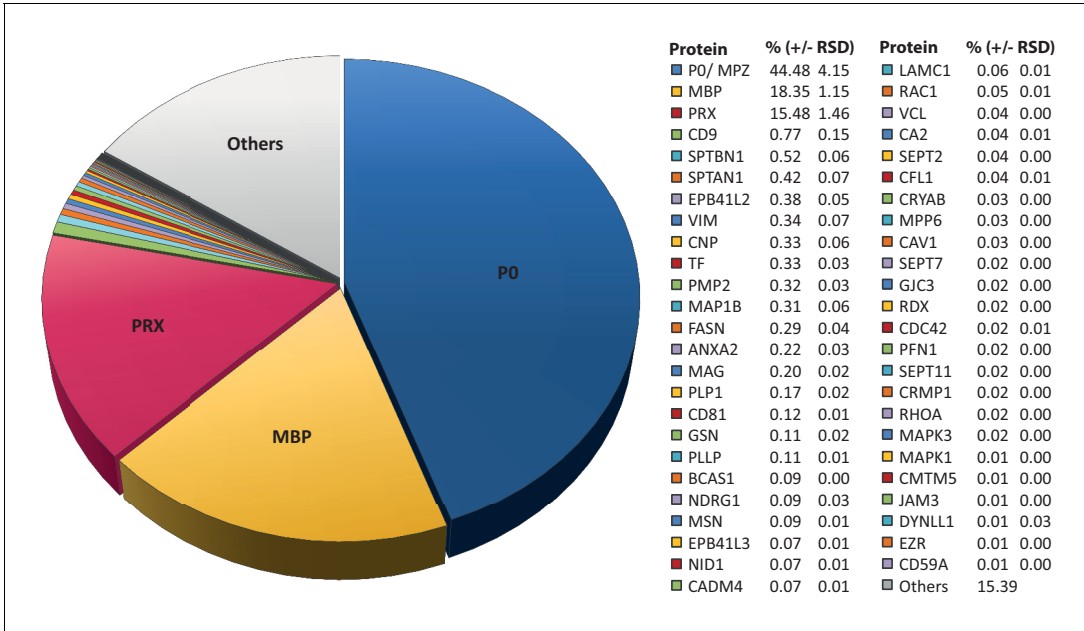

| Protein | % (+/- RSD) | | Protein | % (+/- RSD) | |
|---|---|---|---|---|---|
| P0/ MPZ | 44.48 | 4.15 | LAMC1 | 0.06 | 0.01 |
| MBP | 18.35 | 1.15 | RAC1 | 0.05 | 0.01 |
| PRX | 15.48 | 1.46 | VCL | 0.04 | 0.00 |
| CD9 | 0.77 | 0.15 | CA2 | 0.04 | 0.01 |
| SPTBN1 | 0.52 | 0.06 | SEPT2 | 0.04 | 0.00 |
| SPTAN1 | 0.42 | 0.07 | CFL1 | 0.04 | 0.01 |
| EPB41L2 | 0.38 | 0.05 | CRYAB | 0.03 | 0.00 |
| VIM | 0.34 | 0.07 | MPP6 | 0.03 | 0.00 |
| CNP | 0.33 | 0.06 | CAV1 | 0.03 | 0.00 |
| TF | 0.33 | 0.03 | SEPT7 | 0.02 | 0.00 |
| PMP2 | 0.32 | 0.03 | GJC3 | 0.02 | 0.00 |
| MAP1B | 0.31 | 0.06 | RDX | 0.02 | 0.00 |
| FASN | 0.29 | 0.04 | CDC42 | 0.02 | 0.01 |
| ANXA2 | 0.22 | 0.03 | PFN1 | 0.02 | 0.00 |
| MAG | 0.20 | 0.02 | SEPT11 | 0.02 | 0.00 |
| PLP1 | 0.17 | 0.02 | CRMP1 | 0.02 | 0.00 |
| CD81 | 0.12 | 0.01 | RHOA | 0.02 | 0.00 |
| GSN | 0.11 | 0.02 | MAPK3 | 0.02 | 0.00 |
| PLLP | 0.11 | 0.01 | MAPK1 | 0.01 | 0.00 |
| BCAS1 | 0.09 | 0.00 | CMTM5 | 0.01 | 0.00 |
| NDRG1 | 0.09 | 0.03 | JAM3 | 0.01 | 0.00 |
| MSN | 0.09 | 0.01 | DYNLL1 | 0.01 | 0.03 |
| EPB41L3 | 0.07 | 0.01 | EZR | 0.01 | 0.00 |
| NID1 | 0.07 | 0.01 | CD59A | 0.01 | 0.00 |
| CADM4 | 0.07 | 0.01 | Others | 15.39 | |

**Figure 2.** Relative abundance of peripheral myelin proteins. MS[E] was used to identify and quantify proteins in myelin purified from the sciatic nerves of wild-type mice at P21; their relative abundance is given as percent with relative standard deviation (% +/- RSD). Note that known myelin proteins constitute >80% of the total myelin protein; proteins not previously associated with myelin constitute <20%. Mass spectrometric quantification based on 3 biological replicates per genotype with 4 technical replicates each (see *Figure 1—source data 1*).

large proteins into gels. The present MS[E]-based quantification of myelin proteins also extends beyond and partially adjusts an earlier mass spectrometric approach (*Patzig et al., 2011*). Indeed, the current approach identified and quantified more myelin proteins, probably owing to improved protein solubilization during sample preparation and higher dynamic range of the used mass spectrometer. By MS[E], known myelin proteins (*Table 1*) collectively constitute over 85% of the total myelin protein (*Figure 2*) while proteins not yet associated with myelin account for the remaining 15% of the total myelin protein.

## Comprehensive compendium and comparison to the transcriptome

To systematically elucidate the developmental abundance profiles of the transcripts that encode peripheral myelin proteins (*Figure 3*), we used our combined proteome inventory of peripheral myelin (*Figure 1—source data 1*) to filter mRNA abundance data of all genes expressed in sciatic nerves. By this strategy, *Figure 3* displays only those transcripts of which the protein product was identified in peripheral myelin rather than all transcripts in the nerve, thereby discriminating myelin-related mRNAs from other mRNAs such as those present in peripheral axons, fibroblasts, immune cells etc. In this assessment we additionally included PMP22 although it was not detected by MS as well as 45 proteins exclusively identified by LC-MS of myelin separated by SDS-PAGE (*Figure 1—source data 1*). For mRNA abundance profiles, we exploited a recently established RNA sequencing analysis (RNA-Seq; platform Illumina HiSeq 2000) of sciatic nerves dissected form wild type Sprague Dawley rats at embryonic day 21 (E21), P6, P18 and 6 months (*Fledrich et al., 2018*). RNA-Seq provides reliable information about the relative abundance of all significantly expressed genes and is thus not limited to those represented on the previously used oligonucleotide microarrays (*Patzig et al., 2011*). The raw data (accessible under GEO accession number GSE115930) were normalized (*Figure 3—source data 1*) and standardized. When comparing the proteome and transcriptome datasets, significant mRNA abundance was detected for all 1046 transcripts for which an unambiguous unique gene identifier was found (*Figure 3*). 126 transcripts displayed developmentally unchanged abundance levels, that is, abundance changes below a threshold of 10% coefficient of variation (*Figure 3B*; *Figure 3—source data 1*).

**Table 1.** Known myelin proteins in the myelin proteome.

Proteins mass-spectrometrically identified in peripheral myelin are compiled according to availability of prior references as myelin proteins. Given are the official gene name, one selected reference, the number of transmembrane domains (TMD) and the mRNA abundance profile cluster (see *Figure 3*).

| Protein name | Gene | Reference | TMD | Cluster |
|---|---|---|---|---|
| 2-hydroxyacylsphingosine 1-beta-galactosyltransferase | Ugt8 | Bosio et al., 1996 | 2 | P6-up |
| Syntrophin α1 | Snta1 | Fuhrmann-Stroissnigg et al., 2012 | - | P18-up |
| Annexin A2 | Anxa2 | Hayashi et al., 2007 | - | Descending |
| Band 4.1 protein B/4.1B | Epb41l3 | Ivanovic et al., 2012 | - | Descending |
| Band 4.1 protein G/4.1G | Epb41l2 | Ohno et al., 2006 | - | P6-up |
| Breast carcinoma-amplified sequence 1 | Bcas1 | Ishimoto et al., 2017 | - | P6-up |
| Cadherin 1/E-Cadherin | Cdh1 | Fannon et al., 1995 | 1 | P18-up |
| Carbonic anhydrase 2 | Ca2 | Cammer and Tansey, 1987 | - | Descending |
| Catenin α1 | Ctnna1 | Murata et al., 2006 | - | U-shaped |
| Catenin ß1 | Ctnnb1 | Fannon et al., 1995 | - | Descending |
| Caveolin 1 | Cav1 | Mikol et al., 2002 | 1 | P18-up |
| CD9, tetraspanin 29 | Cd9 | Ishibashi et al., 2004 | 4 | P18-p |
| CD59A | Cd59a | Funabashi et al., 1994 | 1 | P18-up |
| CD47, integrin-associated signal transducer | Cd47 | Gitik et al., 2011 | 5 | P6-up |
| CD81, tetraspanin 28 | Cd81 | Ishibashi et al., 2004 | 4 | P18-up |
| CD82, tetraspanin 27 | Cd82 | Chernousov et al., 2013 | 4 | P18-up |
| CD151, tetraspanin 24 | Cd151 | Patzig et al., 2011 | 4 | P18-up |
| Cell adhesion molecule 4/NECL4 | Cadm4 | Spiegel et al., 2007 | 1 | P6-up |
| Cell division control protein 42 | Cdc42 | Benninger et al., 2007 | - | P6-up |
| Cell surface glycoprotein MUC18 | Mcam | Shih et al., 1998 | 1 | Descending |
| Ciliary neurotrophic factor | Cntf | Rende et al., 1992 | - | Late-up |
| CKLF-like MARVEL TMD-containing 5 | Cmtm5 | Patzig et al., 2011 | 4 | P6-up |
| Claudin-19 | Cldn19 | Miyamoto et al., 2005 | 4 | P6-up |
| Cofilin 1 | Cfl1 | Sparrow et al., 2012 | - | Descending |
| Crystallin α2 | Cryab | D'Antonio et al., 2006 | - | P18-up |
| Cyclic nucleotide phosphodiesterase | Cnp | Matthieu et al., 1980 | - | P6-up |
| Sarcoglycan δ | Sgcd | Cai et al., 2007 | 1 | Late-up |
| Dihydropyrimidinase related protein 1 | Crmp1 | D'Antonio et al., 2006 | - | Descending |
| Disks large homolog 1 | Dlg1 | Cotter et al., 2010 | - | Descending |
| Dynein light chain 1 | Dynll1 | Myllykoski et al., 2018 | - | P6-up |
| Dystroglycan | Dag1 | Yamada et al., 1994 | 1 | P6-up |
| Dystrophin/DP116 | Dmd | Cai et al., 2007 | - | P6-up |
| Dystrophin-related protein 2 | Drp2 | Sherman et al., 2001 | - | P18-up |
| E3 ubiquitin-protein ligase NEDD4 | Nedd4 | Liu et al., 2009 | - | Descending |
| Ezrin | Ezr | Scherer et al., 2001 | - | P6-up |
| Fatty acid synthase | Fasn | Salles et al., 2002 | - | P6-up |
| Flotillin 1 | Flot1 | Lee et al., 2014 | - | P18-up |
| Gap junction ß1 protein/Cx32 | Gjb1 | Li et al., 2002 | 4 | P18-up |
| Gap junction γ3 protein/Cx29 | Gjc3 | Li et al., 2002 | 1 | P6-up |
| Gelsolin | Gsn | Gonçalves et al., 2010 | - | Late-up |
| Glycogen synthase kinase 3ß | Gsk3b | Ogata et al., 2004 | - | P6-up |

*Table 1 continued on next page*

*Table 1 continued*

| Protein name | Gene | Reference | TMD | Cluster |
|---|---|---|---|---|
| Integrin α6 | *Itga6* | *Nodari et al., 2008* | 1 | P6-up |
| Integrin αV | *Itgav* | *Chernousov and Carey, 2003* | 1 | Descending |
| Integrin ß1 | *Itgb1* | *Feltri et al., 2002* | 1 | Descending |
| Integrin ß4 | *Itgb4* | *Quattrini et al., 1996* | 2 | P18-up |
| Junctional adhesion molecule C | *Jam3* | *Scheiermann et al., 2007* | 1 | P18-up |
| Laminin α2 | *Lama2* | *Yang et al., 2005* | - | P6-up |
| Laminin α4 | *Lama4* | *Yang et al., 2005* | - | Descending |
| Laminin ß1 | *Lamb1* | *LeBeau et al., 1994* | - | Descending |
| Laminin ß2 | *Lamb2* | *LeBeau et al., 1994* | - | P18-up |
| Laminin γ1 | *Lamc1* | *Chen and Strickland, 2003* | - | Descending |
| Membrane Palmitoylated Protein 6 | *Mpp6* | *Saitoh et al., 2019* | - | P6-up |
| Microtubule-associated protein 1A | *Map1a* | *Fuhrmann-Stroissnigg et al., 2012* | - | P18-up |
| Microtubule-associated protein 1B | *Map1b* | *Fuhrmann-Stroissnigg et al., 2012* | - | P6-up |
| Mitogen-activated protein kinase 1/ERK2 | *Mapk1* | *Mantuano et al., 2015* | - | Descending |
| Mitogen-activated protein kinase 3/ERK1 | *Mapk3* | *Mantuano et al., 2015* | - | P18-up |
| Moesin | *Msn* | *Scherer et al., 2001* | - | Unchanged |
| Monocarboxylate transporter 1 | *Slc16a1* | *Domènech-Estévez et al., 2015* | 11 | P18-up |
| Myelin associated glycoprotein | *Mag* | *Figlewicz et al., 1981* | 1 | P6-up |
| Myelin basic protein | *Mbp* | *Boggs, 2006* | - | P6-up |
| Myelin protein 2 | *Pmp2* | *Trapp et al., 1984* | - | P18-up |
| Myelin protein zero/P0 | *Mpz* | *Giese et al., 1992* | 1 | P6-up |
| Myelin proteolipid protein | *Plp1* | *Garbern et al., 1997* | 4 | P6-up |
| Myotubularin-related protein 2 | *Mtmr2* | *Bolino et al., 2004* | - | P6-up |
| Noncompact myelin-associated protein | *Ncmap* | *Ryu et al., 2008* | 1 | P18-up |
| NDRG1, N-myc downstream regulated | *Ndrg1* | *Berger et al., 2004* | - | P18-uP |
| Neurofascin | *Nfasc* | *Tait et al., 2000* | 2 | P18-up |
| Nidogen 1 | *Nid1* | *Lee et al., 2007* | - | Descending |
| P2X purinoceptor 7 | *P2r×7* | *Faroni et al., 2014* | - | P6-up |
| Paxillin | *Pxn* | *Fernandez-Valle et al., 2002* | - | P6-up |
| Periaxin | *Prx* | *Gillespie et al., 1994* | - | P6-up |
| Plasmolipin | *Pllp* | *Bosse et al., 2003* | 4 | P18-up |
| Profilin 1 | *Pfn1* | *Montani et al., 2014* | - | Descending |
| Lin-7 homolog C | *Lin7c* | *Saitoh et al., 2017* | - | P6-up |
| Rac1 | *Rac1* | *Benninger et al., 2007* | - | U-Shaped |
| Radixin | *Rdx* | *Scherer et al., 2001* | - | Descending |
| RhoA | *Rhoa* | *Brancolini et al., 1999* | - | U-Shaped |
| Septin 2 | *Sept2* | *Buser et al., 2009* | - | Descending |
| Septin 7 | *Sept7* | *Buser et al., 2009* | - | U-Shaped |
| Septin 8 | *Sept8* | *Patzig et al., 2011* | - | P18-up |
| Septin 9 | *Sept9* | *Patzig et al., 2011* | - | P6-up |
| Septin 11 | *Sept11* | *Buser et al., 2009* | - | Descending |
| Sirtuin 2, NAD-dependent deacetylase | *Sirt2* | *Werner et al., 2007* | - | P18-up |
| Spectrin alpha chain, non-erythrocytic 1 | *Sptan1* | *Susuki et al., 2018* | - | P18-up |
| Spectrin beta chain, non-erythrocytic 1 | *Sptbn1* | *Susuki et al., 2018* | - | P18-up |

*Table 1 continued on next page*

*Table 1 continued*

| Protein name | Gene | Reference | TMD | Cluster |
|---|---|---|---|---|
| Tight junction protein ZO-1 | *Tjp1* | **Poliak et al., 2002** | - | P6-up |
| Tight junction protein ZO-2 | *Tjp2* | **Poliak et al., 2002** | - | P6-up |
| Transferrin | *Tf* | **Lin et al., 1990** | 2 | Late-up |
| Vimentin | *Vim* | **Triolo et al., 2012** | - | Unchanged |
| Vinculin | *Vcl* | **Beppu et al., 2015** | - | Descending |

By fuzzy c-means clustering, those 920 transcripts that showed developmental abundance changes were grouped into 5 clusters (*Figure 3A*; *Figure 3—source data 1*). Among those, one cluster corresponds to an mRNA-abundance peak coinciding with an early phase of myelin biogenesis (cluster 'P6-UP'), which includes the highest proportion of known myelin proteins (*Table 1*) such as MPZ/P0, MBP, PRX, cyclic nucleotide phosphodiesterase (CNP), fatty acid synthase (FASN), myelin-associated glycoprotein (MAG), proteolipid protein (PLP/DM20), cell adhesion molecule-4 (CADM4/NECL4), connexin-29 (GJC3), claudin-19 (CLDN19) and CKLF-like MARVEL-transmembrane domain containing protein-5 (CMTM5). However, many known myelin proteins clustered together according to their mRNA-abundance peak coinciding with a later phase of myelination (cluster 'P18-UP'), including peripheral myelin protein 2 (PMP2), tetraspanin-29 (CD9), tetraspanin-28 (CD81), connexin-32 (GJB1), plasmolipin (PLLP), junctional adhesion molecule-3 (JAM3), CD59 and dystrophin-related protein-2 (DRP2). The proportion of known myelin proteins was lower in the clusters corresponding to mRNA-abundance peaks in adulthood (clusters 'late-UP', 'U-shaped'). Yet, a considerable number of transcripts displayed abundance peaks at the embryonic time-point (cluster 'Descending'), including carbonic anhydrase 2 (CA2), cofilin-1 CFL1), tubulin beta-4 (TUBB4b) and band 4.1-protein B (EPB41L3). Generalized, the clusters were roughly similar when comparing previous oligonucleotide microarray analysis of mouse sciatic nerves (*Patzig et al., 2011*) and the RNA-Seq analysis of rat sciatic nerves (this study); yet, the latter provides information on a larger number of genes and with a higher level of confidence. Together, clustering of mRNA abundance profiles allows categorizing peripheral myelin proteins into developmentally co-regulated groups.

When systematically assessing the proteins identified in myelin by gene ontology (GO)-term analysis, the functional categories over-represented in the entire myelin proteome included cell adhesion, cytoskeleton and extracellular matrix (labeled in turquoise in *Figure 4*). When analyzing the clusters of developmentally co-expressed transcripts (from *Figure 3*), proteins associated with the lipid metabolism were particularly enriched in the P6-UP and P18-UP clusters, while those associated with the extracellular matrix (ECM) were over-represented in the U-shaped and Descending clusters (*Figure 4*). For comparison, known myelin proteins (*Table 1*) were over-represented in the P6-UP and P18-UP clusters (*Figure 4*). Together, our proteome dataset provides comprehensive in-depth coverage of the protein constituents of peripheral myelin purified from the sciatic nerves of wild type mice, and comparison to the transcriptome allows identifying developmentally co-regulated and functional groups of myelin proteins. Our data thus supply a solid resource for the molecular characterization of myelin and for discovering functionally relevant myelin proteins.

## Neuropathy genes encoding myelin proteins

Heritable neuropathies can be caused by mutations affecting genes preferentially expressed in neurons, Schwann cells or both (*Rossor et al., 2013*; *Pareyson and Marchesi, 2009*; *Baets et al., 2014*; *Brennan et al., 2015*). To systematically assess which neuropathy-causing genes encode peripheral myelin proteins, we compared our myelin proteome dataset with a current overview about disease genes at the NIH National Library of Medicine at https://ghr.nlm.nih.gov/condition/charcot-marie-tooth-disease#genes. Indeed, 31 myelin proteins were identified to be encoded by a proven neuropathy gene (*Table 2*), a considerable increase compared to eight disease genes found in a similar previous approach (*Patzig et al., 2011*). Notably, this increase is owing to both the larger size of the current myelin proteome dataset (*Figure 1E*) and the recent discovery of numerous neuropathy genes by the widespread application of next generation sequencing.

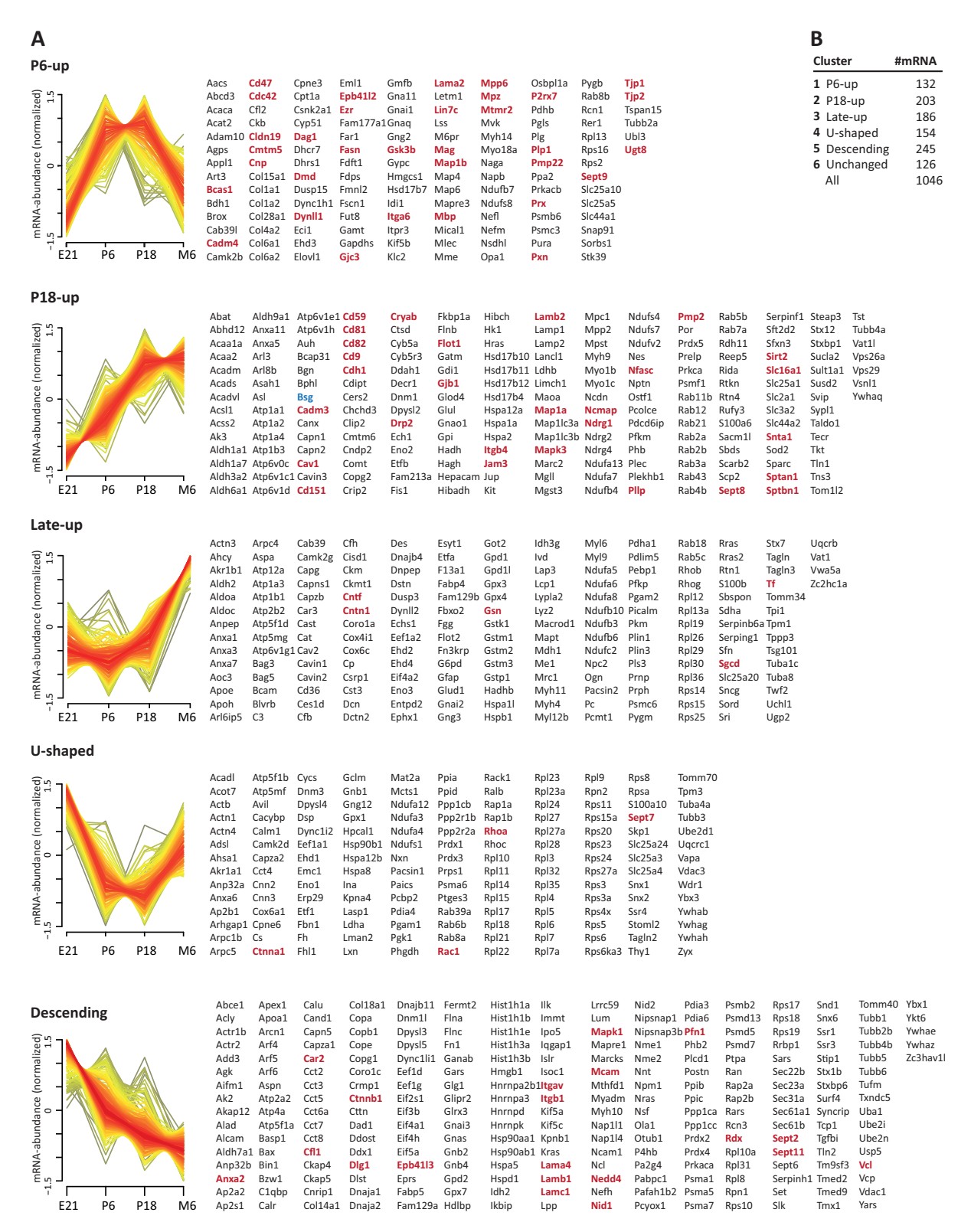

**Figure 3.** Developmental mRNA abundance profiles of myelin-associated genes. (**A**) K-means clustering was performed for the mRNA profiles of those 1046 proteins in our myelin proteome inventory for which significant mRNA expression was found by RNA-Seq in the sciatic nerve of rats dissected at ages E21, P6, P18 and 6 months (M6). Note that this filtering strategy allows to selectively display the developmental abundance profiles of those transcripts that encode myelin-associated proteins rather than of all transcripts present in the nerve. Standardized mRNA abundance profiles are shown

*Figure 3 continued on next page*

*Figure 3 continued*

(n = 4 biological replicates per age). Known myelin genes are displayed in red. For comparison, *Pmp22* mRNA was included although the small tetraspan protein PMP22 was not mass spectrometrically identified due to its unfavorable distribution of tryptic cleavage sites. Normalized counts for all mRNAs including those displaying developmentally unchanged abundance are provided in *Figure 3—source data 1*. (B) Numbers of mRNAs per cluster.

The online version of this article includes the following source data for figure 3:

**Source data 1.** Normalized developmental mRNA abundance data → sheet 1: normalized values for all individual 4 biological replicates per age → sheet 2: normalized values for biological replicates averaged to give mean per age.

## Pathological proteomic profile of peripheral myelin in a neuropathy model

The results presented thus far were based on analyzing myelin of healthy wild type mice; yet we also sought to establish a straightforward method to systematically assess myelin diversity, as exemplified by alterations in a pathological situation. As a model we chose mice carrying a homozygous deletion of the periaxin gene ($Prx^{-/-}$) (*Court et al., 2004*; *Gillespie et al., 2000*). Periaxin (PRX) is the third-most abundant peripheral myelin protein (*Figure 2*) and scaffolds the dystroglycan complex in Schwann cells. $Prx^{-/-}$ mice represent an established model of Charcot-Marie-Tooth disease type 4F (*Guilbot et al., 2001*; *Berger et al., 2006*; *Marchesi et al., 2010*). Aiming to assess the myelin proteome, we purified myelin from pools of sciatic nerves dissected from $Prx^{-/-}$ and control mice at P21. Upon SDS-PAGE separation and silver staining the band patterns appeared roughly similar (*Figure 5A*), with the most obvious exception of the absence of the high-molecular weight band constituted by periaxin in $Prx^{-/-}$ myelin. Yet, several other bands also displayed genotype-dependent differences in intensity. As expected, PRX was also undetectable by MS$^E$ in $Prx^{-/-}$ myelin, in which

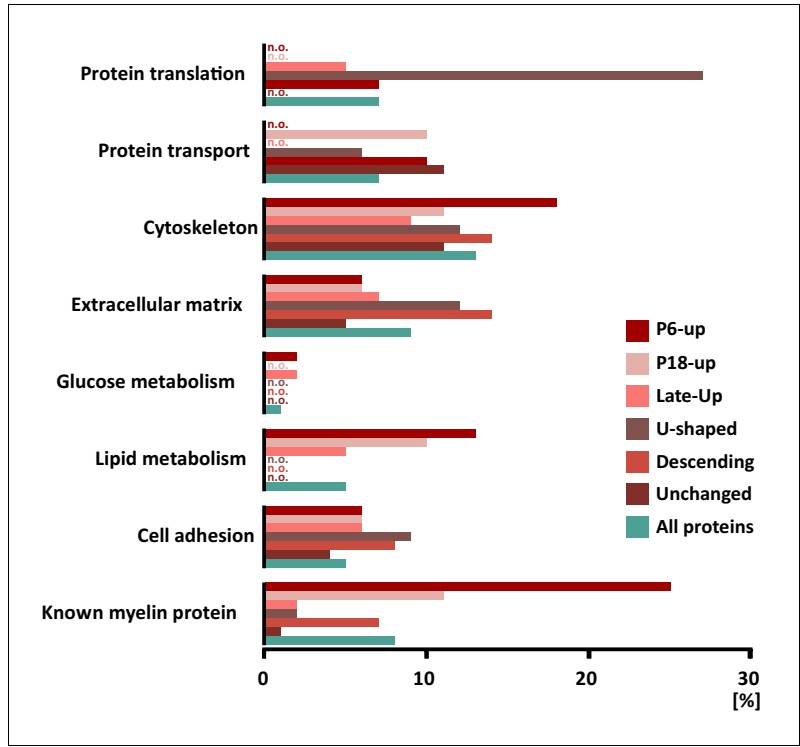

**Figure 4.** Categorization of annotated protein functions. All proteins identified in peripheral myelin by UDMS$^E$ (turquoise) and the respective developmental expression clusters (*Figure 3*; shades of red) were analyzed for overrepresented functional annotations using gene ontology (GO) terms. The graph displays the percentage of proteins in each cluster that were annotated with a particular function. For comparison, known myelin proteins were annotated. n.o., not over-represented.

**Table 2.** Peripheral myelin proteins identified in PNS myelin involved in neuropathological diseases.

Proteins mass-spectrometically identified in peripheral myelin were analyzed regarding the involvement of the ortholog human gene in neuropathological diseases. PMP22 was added, though it was not identified by MS analyses due to its unfavorable distribution of tryptic cleavage sites. CMT, Charcot-Marie-Tooth disease; DHMN, distal hereditary motor neuropathy; DI-CMTC, dominant intermediate CMTC; DFN, X-linked deafness; HMN, hereditary motor neuropathy; HSAN, hereditary sensory and autonomic neuropathy; HNA, hereditary sensory and autonomic neuropathy; OMIM, Online Mendelian Inheritance in Man; PHARC, polyneuropathy, hearing loss, ataxia, retinitis pigmentosa and cataract; SCA, spinocerebellar ataxia; SPG, spastic paraplegia.

| Protein name | Gene name | OMIM# | Gene locus | Neuropathy |
|---|---|---|---|---|
| Monoacylglycerol lipase ABHD12 | ABHD12 | 613599 | 20p11.21 | Pharc |
| Apoptosis-inducing factor 1 | AIFM1 | 300169 | Xq26.1 | CMTX4, DFNX5 |
| Na+/K+ -transporting ATPase α1 | ATP1A1 | 182310 | 1p13.1 | CMT2DD |
| Cytochrome c oxidase subunit 6A1 | COX6A1 | 602072 | 12q24.31 | CMTRID |
| Dystrophin-related protein 2 | DRP2 | 300052 | Xq22.1 | CMTX |
| Dynactin subunit 1 | DCTN1 | 601143 | 2p13.1 | DHMN7B |
| Dynamin 2 | DNM2 | 602378 | 19p13.2 | CMT2M, CMTDIB |
| Cytoplasmic dynein 1 heavy chain 1 | DYNC1H1 | 600112 | 14q32.31 | CMT20, SMALED1 |
| E3 SUMO-protein ligase | EGR2 | 129010 | 10q21.3 | CMT1D, CMT3, CMT4E |
| Glycine-tRNA ligase | GARS (Gart) | 600287 | 7p14.3 | CMT2D, HMN5A |
| Gap junction ß1 protein/Cx32 | GJB1 | 304040 | Xq13.1 | CMTX1 |
| Guanine nucleotide-binding protein ß4 | GNB4 | 610863 | 3q26.33 | CMTDIF |
| Histidine triad nucleotide-binding protein 1 | HINT1 | 601314 | 5q23.3 | NMAN |
| Hexokinase 1 | HK1 | 142600 | 10q22.1 | CMT4G |
| Heat shock protein ß1 | HSPB1 | 602195 | 7q11.23 | CMT2F, DHMN2B |
| Kinesin heavy chain isoform 5A | KIF5A | 602821 | 12q13.3 | SPG10 |
| Prelamin A/C | LMNA | 150330 | 1q22 | CMT2B1 |
| Neprilysin | MME | 120520 | 3q25.2 | CMT2T, SCA43 |
| Myelin protein zero/P0 | MPZ | 159440 | 1q23.3 | CHN2,CMT1B, CMT2I, CMT2J,CMT3, CMTDID, Roussy-Levy syndrome |
| Myotubularin-related protein 2 | MTMR2 | 603557 | 11q21 | CMT4B1 |
| Alpha-N-acetylglucosaminidase | NAGLU (NAGA) | 609701 | 17q21.2 | CMT2V |
| NDRG1, N-myc downstream regulated | NDRG1 | 605262 | 8q24.22 | CMT4D |
| Neurofilament heavy polypeptide | NEFH | 162230 | 22q12.2 | CMT2CC |
| Neurofilament light polypeptide | NEFL | 162280 | 8p21.2 | CMT2E, CMT1F, CMTDIG |
| Peripheral myelin protein 2 | PMP2 | 170715 | 8q21.13 | CMT1G |
| Peripheral myelin protein 22 | PMP22 | 601907 | 17p12 | CMT1A, CMT1E, CMT3, HNPP, Roussy-Levy syndrome |
| Ribose-phosphate pyrophosphokinase 1 | PRPS1 | 311850 | Xq22.3 | Arts syndrome, CMTX5, DFNX1 |
| Periaxin | PRX | 605725 | 19q13.2 | CMT4F, CMT3 |
| Ras-related protein Rab 7a | RAB7A | 602298 | 3q21.3 | CMT2B |
| Septin 9 | SEPT9 | 604061 | 17q25.3 | HNA |
| Transitional ER-ATPase | VCP | 601023 | 9p13.3 | CMT2Y |
| Tryptophan-tRNA ligase, cytoplasmic | WARS | 191050 | 14q32.32 | HMN9 |
| Tyrosine-tRNA ligase, cytoplasmic | YARS | 603623 | 1p35.1 | DI-CMTC |

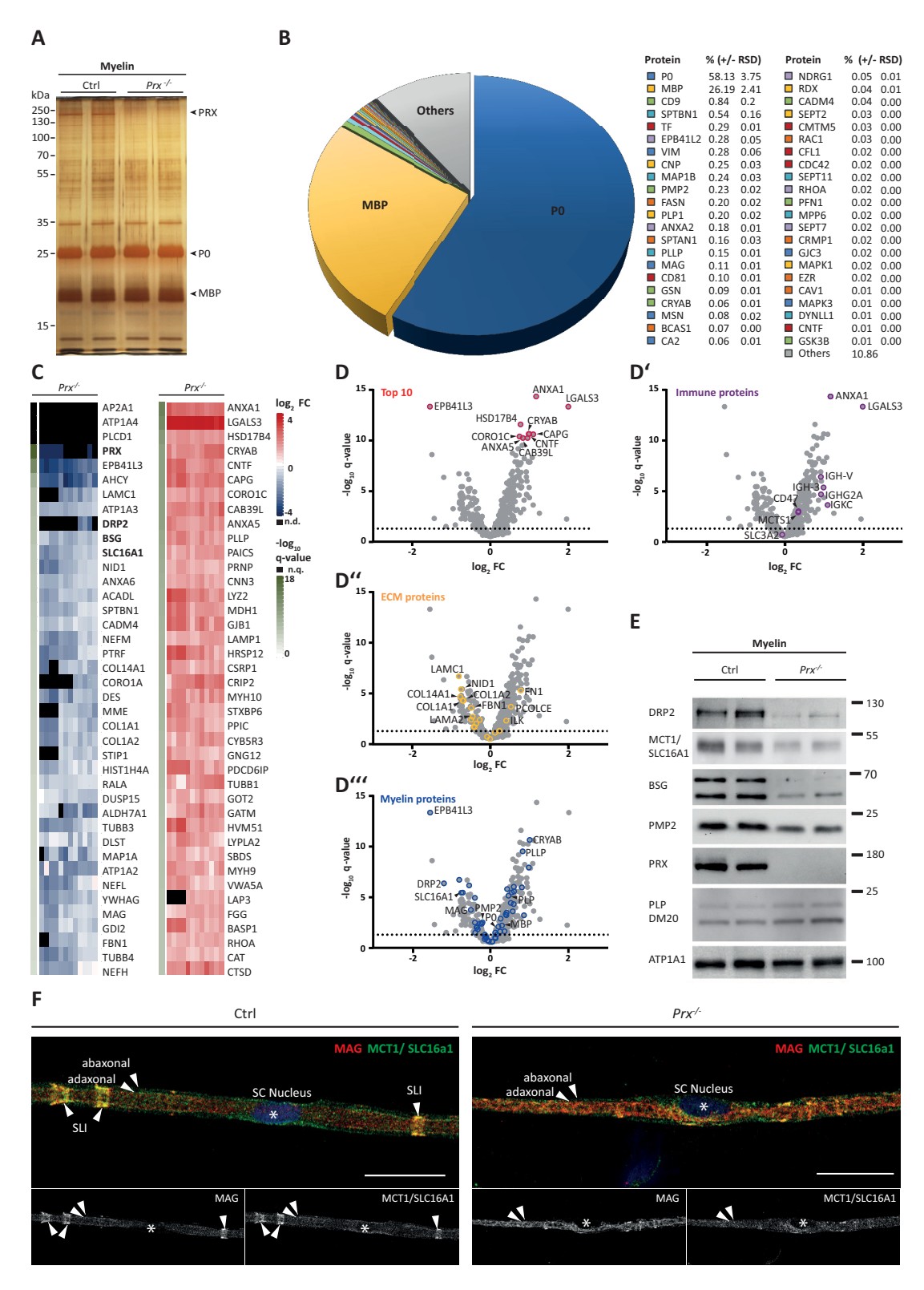

**Figure 5.** Molecular analysis of myelin in the *Prx*[-/-] mouse model of CMT4F. (**A**) Myelin purified from sciatic nerves dissected from *Prx*[-/-] and control mice at P21 was separated by SDS-PAGE (0.5 μg protein load) and proteins were visualized by silver staining. Bands constituted by the most abundant myelin proteins (MPZ/P0, MBP, PRX) are annotated. Note that no band constituted by PRX was detected in *Prx*[-/-] myelin and that several other bands also display genotype-dependent differences in intensity. Gel shows n = 2 biological replicates representative of n = 3 biological replicates. (**B**) The

*Figure 5 continued on next page*

*Figure 5 continued*

relative abundance of proteins in myelin purified from *Prx*<sup>-/-</sup> sciatic nerves as quantified by MS<sup>E</sup> is given as percent with relative standard deviation (% +/- RSD). Note the increased relative abundance of MPZ/P0 and MBP compared to wild-type myelin (see *Figure 2*) when PRX is lacking. Mass spectrometric quantification based on 3 biological replicates with 4 technical replicates each (see *Figure 5—source data 1*). (C,D) Differential proteome analysis by DRE-UDMS<sup>E</sup> of myelin purified from *Prx*<sup>-/-</sup> and wild-type mice. Mass spectrometric quantification based on 3 biological replicates per genotype with 4 technical replicates each (see *Figure 5—source data 2*). (C) Top 40 proteins of which the abundance is reduced (blue) or increased (red) in peripheral myelin purified from *Prx*<sup>-/-</sup> compared to wild-type mice with the highest level of significance according to the -log<sub>10</sub> transformed q-value (green). In the heatmaps, each horizontal line corresponds to the fold-change (FC) of a distinct protein compared to its average abundance in wild-type myelin plotted on a log<sub>2</sub> color scale. Heatmaps display 12 replicates, that is 3 biological replicates per genotype with 4 technical replicates each. (D-D''') Volcano plots representing genotype-dependent quantitative myelin proteome analysis. Data points represent quantified proteins in *Prx*<sup>-/-</sup> compared to wild-type myelin and are plotted as the log2-transformed fold-change (FC) on the x-axis against the -log10-transformed q-value on the y-axis. Stippled lines mark a -log10-transformed q-value of 1.301, reflecting a q-value of 0.05 as significance threshold. Highlighted are the datapoints representing the Top 10 proteins displaying highest zdist values (Euclidean distance between the two points (0,0) and (x,y) with x = log2(FC) and y = -log10(q-value) (red circles in D), immune-related proteins (purple circles in D'), proteins of the extracellular matrix (ECM; yellow circles in D'') and known myelin proteins (blue circles in D'''). n.d., not detected; n.q., no q-value computable due to protein identification in one genotype only. Also see *Figure 5—figure supplement 1*. (E) Immunoblot of myelin purified from *Prx*<sup>-/-</sup> and control sciatic nerves confirms the reduced abundance of DRP2, SLC16A1/MCT1, BSG and PMP2 in *Prx*<sup>-/-</sup> myelin, as found by differential DRE-UDMS<sup>E</sup> analysis (in C,D). PRX was detected as genotype control; PLP/DM20 and ATP1A1 serve as markers. Blot shows n = 2 biological replicates per genotype. (F) Teased fiber preparations of sciatic nerves dissected from *Prx*<sup>-/-</sup> and control mice immunolabelled for MAG (red) and SLC16A1 (green). Note that SLC16A1 co-distributes with MAG in Schmidt-Lanterman incisures (SLI) in control but not in *Prx*<sup>-/-</sup> nerves, in accordance with the reduced abundance of SLC16A1 in *Prx*<sup>-/-</sup> myelin (C–E). Also note that, in *Prx*<sup>-/-</sup> myelin, SLI were largely undetectable by MAG immunolabeling.

The online version of this article includes the following source data and figure supplement(s) for figure 5:

**Source data 1.** Label-free quantification of proteins in PNS myelin fractions from *Prx*<sup>-/-</sup> mice by MS<sup>E</sup> Identification and quantification data of detected myelin-associated proteins.

**Source data 2.** Label-free quantification of proteins in PNS myelin fractions from WT and *Prx*<sup>-/-</sup> mice by DRE-UDMS<sup>E</sup> Identification and quantification data of detected myelin-associated proteins by DRE-UDMS<sup>E</sup>.

**Figure supplement 1.** Clustered heatmap of Pearson's correlation coefficients for protein abundance comparing genotypes.

most of the total myelin protein was constituted by MPZ/P0 and MBP (*Figure 5B*; *Figure 5—source data 1*).

Upon differential analysis by DRE-UDMS<sup>E</sup> (*Figure 5—source data 2*), multiple proteins displayed genotype-dependent differences as visualized in a heatmap displaying those 40 proteins of which the abundance was reduced or increased with the highest statistical significance in *Prx*<sup>-/-</sup> compared to control myelin (*Figure 5C*). For example, the abundance of the periaxin-associated dystrophin-related protein 2 (DRP2) was strongly reduced in *Prx*<sup>-/-</sup> myelin, as previously shown by immunoblotting (*Sherman et al., 2001*). Notably, the abundance of multiple other proteins was also significantly reduced in *Prx*<sup>-/-</sup> myelin, including the extracellular matrix protein laminin C1 (LAMC1; previously termed LAMB2), the laminin-associated protein nidogen (NID1), Ig-like cell adhesion molecules (CADM4, MAG), the desmosomal junction protein desmin (DES), cytoskeletal and cytoskeleton-associated proteins (EPB41L3, MAP1A, CORO1A, SPTBN1, various microtubular and intermediate filament monomers), the monocarboxylate transporter MCT1 (also termed SLC16A1) and the MCT1-associated (*Philp et al., 2003*) immunoglobulin superfamily protein basigin (BSG, also termed CD147). On the other hand, proteins displaying the strongest abundance increase in *Prx*<sup>-/-</sup> myelin included immune-related proteins (LGALS3, LYZ2, CTSD), cytoskeletal and cytoskeleton-associated proteins (CAPG, CORO1C, CNN3, several myosin heavy chain subunits), peroxisomal enzymes (CAT, HSD17B4, MDH1) and known myelin proteins (PLLP/plasmolipin, CRYAB, GJB1/CX32). For comparison, the abundance of the marker proteolipid protein (PLP/DM20) (*Patzig et al., 2016a*) and the periaxin-associated integrin beta-4 (ITGB4) (*Raasakka et al., 2019*) in myelin was unaltered in *Prx*<sup>-/-</sup> myelin. Together, differential proteome analysis finds considerably more proteins and protein groups to be altered in *Prx*<sup>-/-</sup> myelin than previously known (*Figure 5C,D–D'''*), probably reflecting the complex pathology observed in this model (*Court et al., 2004*; *Gillespie et al., 2000*).

The monocarboxylate transporter MCT1/SLC16A1 expressed by myelinating oligodendrocytes (*Lee et al., 2012*; *Fünfschilling et al., 2012*) and Schwann cells (*Domènech-Estévez et al., 2015*; *Morrison et al., 2015*) has been proposed to supply lactate or other glucose breakdown products to axons, in which they may serve as substrate for the mitochondrial production of ATP (*Morrison et al., 2013*; *Saab et al., 2013*; *Rinholm and Bergersen, 2014*). In this respect it was

striking to find the abundance of MCT1 significantly reduced in peripheral myelin when PRX is lacking (*Figure 5C*), a result that we were able to confirm by immunoblotting (*Figure 5E*) and immuno-labeling of teased fiber preparations of sciatic nerves (*Figure 5F*). Notably, reduced expression of MCT1 in *Slc16a1*$^{+/-}$ mice impairs axonal integrity at least in the CNS (*Lee et al., 2012*; *Jha et al., 2020*). The reduced abundance of MCT1 thus represents an interesting novel facet of the complex pathology in *Prx*$^{-/-}$ mice. Considering that the integrity of peripheral axons may be impaired in *Prx*$^{-/-}$ mice, we assessed their quadriceps nerves. Indeed, *Prx*$^{-/-}$ mice displayed reduced axonal diameters, a progressively reduced total number of axons and a considerable number of myelin whorls lacking a visible axon (*Figure 6*), indicative of impaired axonal integrity (*Edgar et al., 2009*). Yet we note that molecular or neuropathological features other than the reduced abundance of MCT1 probably also contribute to the axonopathy in *Prx*$^{-/-}$ mice.

Together, gel-free, label free proteome analysis provides a cost- and time-efficient method that provides an accurate, sensitive tool to gain systematic insight into the protein composition of healthy peripheral myelin and its alterations in pathological situations. Indeed, gel-free proteome analysis is particularly powerful and comprehensive compared to 2D-DIGE; the workflow presented here appears readily applicable to other neuropathy models, thereby promising discovery of relevant novel features of their neuropathology.

## Discussion

We used gel-free, label-free quantitative mass spectrometry to assess the protein composition of myelin biochemically purified from the sciatic nerves of wild-type mice, thereby establishing a straightforward and readily applicable workflow to approach the peripheral myelin proteome. The key to comprehensiveness was to combine the strengths of three data acquisition modes, that is, MS$^{E}$ for correct quantification of high-abundant proteins, UDMS$^{E}$ for deep quantitative proteome coverage including low-abundant proteins and DRE-UDMS$^{E}$ for differential analysis. We suggest that DRE-UDMS$^{E}$ provides a good compromise between dynamic range, identification rate and instrument run time for routine differential myelin proteome profiling as a prerequisite for a molecular understanding of myelin (patho)biology. We have also integrated the resulting compendium with RNA-Seq-based mRNA abundance profiles in peripheral nerves and neuropathy disease loci. Beyond providing the largest peripheral myelin proteome dataset thus far, the workflow is appropriate to serve as starting point for assessing relevant variations of myelin protein composition, for example, in different nerves, ages, species and in pathological conditions. The identification of numerous pathological alterations of myelin protein composition in the *Prx*$^{-/-}$ neuropathy model indicates that the method is well suited to assess such diversity.

Aiming to understand nervous system function at the molecular level, multiple 'omics'-scale projects assess the spatio-temporal expression profiles of all mRNAs and proteins in the CNS including oligodendrocytes and myelin (*Zhang et al., 2014*; *Patzig et al., 2016b*; *Sharma et al., 2015*; *Thakurela et al., 2016*; *Marques et al., 2016*). Yet, peripheral nerves are also essential for normal sensory and motor capabilities. Prior approaches to the molecular profiles of Schwann cells and PNS myelin thus far, however, were performed >8 years ago (*Nagarajan et al., 2002*; *Le et al., 2005*; *Ryu et al., 2008*; *Patzig et al., 2011*; *Verheijen et al., 2003*; *Buchstaller et al., 2004*; *D'Antonio et al., 2006*), and the techniques have considerably advanced since. For example, current gel-free, label-free mass spectrometry can simultaneously identify and quantify the vast majority of proteins in a sample, thereby providing comprehensive in depth-information. Moreover, RNA-Seq technology has overcome limitations of the previously used microarrays for characterizing mRNA abundance profiles with respect to the number of represented genes and the suitability of the oligo-nucleotide probes. The present compendium thus provides high confidence with respect to the identification of myelin proteins, their relative abundance and their developmental mRNA expression profiles. This view is supported by the finding that over 80% of the total myelin proteome is constituted by approximately 50 previously known myelin proteins. We believe that the majority of the other identified proteins represent low-abundant myelin-associated constituents in line with the high efficiency of biochemical myelin purification. Doubtless, however, the myelin proteome also comprises contaminants from other cellular sources, underscoring the need of independent validation for establishing newly identified constituents as true myelin proteins.

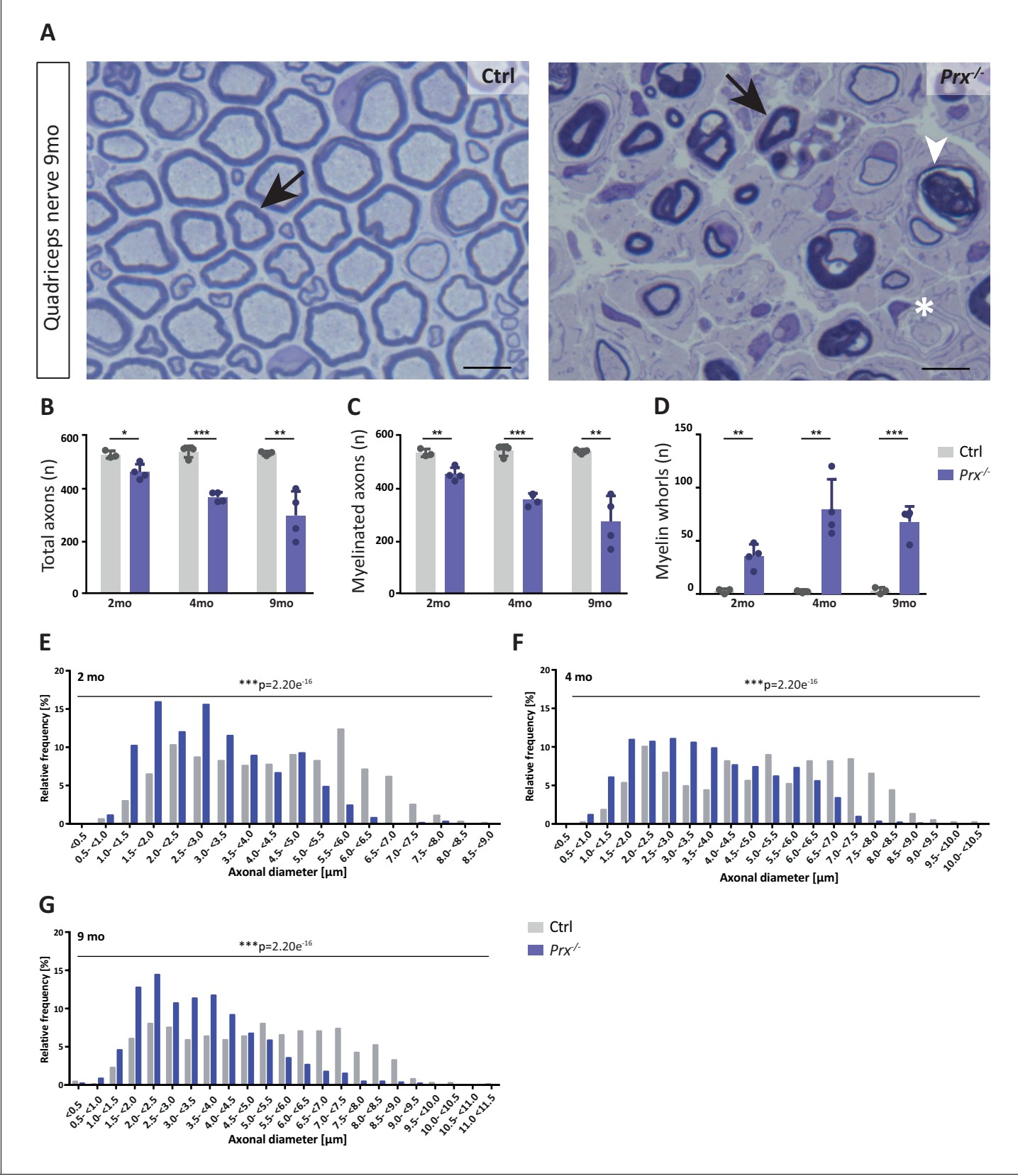

**Figure 6.** Progressive loss and reduced diameters of peripheral axons in *Prx*$^{-/-}$ mice. (**A–D**) Genotype-dependent quantitative assessment of light micrographs of toluidine-stained semi-thin sectioned quadriceps nerves dissected at 2, 4 and 9 months of age reveals progressive loss of peripheral

*Figure 6 continued on next page*

*Figure 6 continued*

axons in *Prx*⁻/⁻ compared to control mice. (**A**) Representative micrographs. Arrows point at myelinated axons; asterisk denotes an unmyelinated axon; arrowhead points at a myelin whorl lacking a recognizable axon. Scale bars, 10 µm. (**B**) Total number of axons per nerve that are not associated with a Remak bundle. (**C**) Total number of myelinated axons per nerve. (**D**) Total number per nerve of myelin whorls that lack a recognizable axon. Mean +/SD, n = 3–4 mice per genotype and age; *p<0.05, **p<0.01, ***p<0.001 by Student's unpaired t-test. (**E–G**) Genotype-dependent assessment of myelinated axons shows a shift toward reduced axonal diameters in quadriceps nerves of *Prx*⁻/⁻ compared to control mice at 2 months (**E**), 4 months (**F**) and 9 months (**G**) of age. Data are presented as frequency distribution with 0.5 µm bin width. ***, p<0.001 by two-sided Kolmogorow-Smirnow test. For precise p-values see methods section.

Do myelin proteins exist that escape identification by standard proteomic approaches? Indeed, some proteins display atypically distributed lysine and arginine residues, which represent the cleavage sites of the commonly used protease trypsin. The tryptic digest of these proteins leads to peptides that are not well suited for chromatographic separation and/or mass spectrometric detection/sequencing, as exemplified by the small hydrophobic tetraspan-transmembrane myelin proteins MAL (*Schaeren-Wiemers et al., 2004*) and PMP22 (*Adlkofer et al., 1995*). We can thus not exclude that additional proteins with atypical tryptic digest patterns exist in peripheral myelin, which would need to be addressed by the use of alternative proteases. Moreover, potent signaling molecules including erbB receptor tyrosine kinases (*Riethmacher et al., 1997*; *Woldeyesus et al., 1999*) and G-protein coupled receptors (GPRs) (*Ackerman et al., 2018*; *Monk et al., 2011*; *Monk et al., 2009*) display exceptionally low abundance. Such proteins may be identified when applying less stringent identification criteria, e.g., by requiring the sequencing of only one peptide per protein. However, lower stringency would also result in identifying false-positive proteins, which we wished to avoid for the purpose of the present compendium. We note that a truly comprehensive spatio-temporally resolved myelin proteome should preferentially also include systematic information about protein isoforms and post-translational modifications, which still poses technical challenges.

Mutations affecting the periaxin (*PRX*) gene in humans cause CMT type 4F (*Guilbot et al., 2001*; *Kabzinska et al., 2006*; *Baránková et al., 2008*; *Tokunaga et al., 2012*); the neuropathology resulting from mutations affecting periaxin has been mainly investigated in the *Prx*⁻/⁻ mouse model. Indeed, *Prx*⁻/⁻ mice display a progressive peripheral neuropathy including axon/myelin-units with abnormal myelin thickness, demyelination, tomaculae, onion bulbs, reduced nerve conduction velocity (*Gillespie et al., 2000*), reduced abundance and mislocalization of the periaxin-associated DRP2 (*Sherman et al., 2001*) and reduced internode length (*Court et al., 2004*). Absence of SLIs (*Gillespie et al., 2000*) and bands of Cajal (*Court et al., 2004*) imply that the non-compact myelin compartments are impaired when PRX is lacking. In the differential analysis of myelin purified from *Prx*⁻/⁻ and control mice we find that the previously reported reduced abundance of DRP2 (*Sherman et al., 2001*) represents one of the strongest molecular changes in the myelin proteome when PRX is lacking. Notably, the reported morphological changes in this neuropathy model (*Sherman et al., 2001*; *Court et al., 2004*; *Gillespie et al., 2000*) go along with alterations affecting the abundance of multiple other myelin-associated proteins, including junctional, cytoskeletal, extracellular matrix and immune-related proteins as well as lipid-modifying enzymes. Thus, the neuropathology in *Prx*⁻/⁻ mice at the molecular level is more complex than previously anticipated. It is striking that the abundance of the monocarboxylate transporter MCT1/SLC16A1 that may contribute to the metabolic supply of lactate from myelinating cells to axons (*Beirowski et al., 2014*; *Domènech-Estévez et al., 2015*; *Kim et al., 2016*; *Gonçalves et al., 2017*; *Stassart et al., 2018*) is strongly reduced in *Prx*⁻/⁻ myelin. Considering that MCT1 in Schwann cells mainly localizes to Schmidt Lanterman incisures (SLI) (*Domènech-Estévez et al., 2015*) and that SLI are largely absent from myelin when PRX is lacking (*Gillespie et al., 2000*), the reduced abundance of MCT1 in *Prx*⁻/⁻ myelin may be a consequence of the impaired myelin ultrastructure. Yet, considering that SLI are part of the cytosolic channels that may represent transport routes toward Schwann cell-dependent metabolic support of myelinated axons, the diminishment of MCT1 may contribute to reduced axonal diameters or axonal loss in *Prx*⁻/⁻ mice, probably in conjunction with other molecular or morphological defects. Together, the in depth-analysis of proteins altered in neuropathy models can contribute to an improved understanding of nerve pathophysiology.

Compared to a previous approach (*Patzig et al., 2011*), the number of proven neuropathy genes of which the encoded protein is mass spectrometrically identified in peripheral myelin has increased

four-fold from eight to 32 in the present study. This reflects both that the number of proteins identified in myelin has approximately doubled and that more neuropathy genes are known due to the common use of genome sequencing. We note that our compendium comprises not only myelin-associated proteins causing (when mutated) demyelinating CMT1 (e.g., MPZ/P0, NEFL, PMP2) or intermediate CMT4 (GDAP1, NDRG1, PRX) but also axonal CMT2 (RAB7, GARS, HSPB1). Yet, the expression of genes causative of CMT2 is not necessarily limited to neurons, as exemplified by the classical myelin protein MPZ/P0. Indeed, a subset of *MPZ*-mutations causes axonal CMT2I or CMT2J (*Gallardo et al., 2009*; *Leal et al., 2014*; *Tokuda et al., 2015*; *Duan et al., 2016*; *Fabrizi et al., 2018*), probably reflecting impaired axonal integrity as consequence of a mutation primarily affecting Schwann cells. We also note that the nuclear EGR2/KROX20 causative of demyelinating CMT1D has not been mass spectrometrically identified in myelin, reflecting that Schwann cell nuclei are efficiently removed during myelin purification.

While morphological analysis of peripheral nerves by light and electron microscopy is routine in numerous laboratories, systematic molecular analysis has been less straightforward. Using the sciatic nerve as a model, we show that systematic assessment of the myelin proteome and the total nerve transcriptome are suited to determine comprehensive molecular profiles in healthy nerves and in myelin-related disorders. Myelin proteome analysis can thus complement transcriptome analysis in assessing development, function and pathophysiology of peripheral nerves.

## Materials and methods

### Mouse models

*Prx^{-/-}* mice (*Gillespie et al., 2000*) were kept on c57Bl/6 background in the animal facility of the University of Edinburgh (United Kingdom). Genotyping was by PCR on genomic DNA using the forward primers 5'-CAGATTTGCT CTGCCCAAGT and 5'-CGCCTTCTAT CGCCTTCTTGAC in combination with reverse primer 5'-ATGCCCTCAC CCACTAACAG. The PCR yielded a 0.5 kb fragment for the wildtype allele and a 0.75 kb product for the mutant allele. The age of experimental animals is given in the figure legends. All animal work conformed to United Kingdom legislation (Scientific Procedures) Act 1986 and to the University of Edinburgh Ethical Review Committee policy; Home Office project license No. P0F4A25E9.

### Myelin purification

A light-weight membrane fraction enriched for myelin was purified from sciatic nerves of mice by sucrose density centrifugation and osmotic shocks as described (*Patzig et al., 2011*; *Erwig et al., 2019a*). Myelin accumulates at the interface between 0.29 and 0.85 M sucrose. *Prx^{-/-}* and wild type control C57Bl/6 mice were sacrificed by cervical dislocation at postnatal day 21 (P21). For each genotype, myelin was purified as three biological replicates (n = 3); each biological replicate representing a pool of 20 sciatic nerves dissected from 10 mice. Protein concentration was determined using the DC Protein Assay Kit (Bio-Rad).

### Filter-aided sample preparation for proteome analysis

Protein fractions corresponding to 10 µg myelin protein were dissolved and processed according to a filter-aided sample preparation (FASP) protocol essentially as previously described for synaptic protein fractions (*Ambrozkiewicz et al., 2018*) and as adapted to CNS myelin (*Erwig et al., 2019a*; *Erwig et al., 2019b*). Unless stated otherwise, all steps were automated on a liquid-handling workstation equipped with a vacuum manifold (Freedom EVO 150, Tecan) by using an adaptor device constructed in-house. Briefly, myelin protein samples were lysed and reduced in lysis buffer (7 M urea, 2 M thiourea, 10 mM DTT, 0.1 M Tris pH 8.5) containing 1% ASB-14 by shaking for 30 min at 37°C. Subsequently, the sample was diluted with ~10 volumes lysis buffer containing 2% CHAPS to reduce the ASB-14 concentration and loaded on centrifugal filter units (30 kDa MWCO, Merck Millipore). After removal of the detergents by washing twice with wash buffer (8 M urea, 10 mM DTT, 0.1 M Tris pH 8.5), proteins were alkylated with 50 mM iodoacetamide in 8 M urea, 0.1 M Tris pH 8.5 (20 min at RT), followed by two washes with wash buffer to remove excess reagent. Buffer was exchanged by washing three times with 50 mM ammonium bicarbonate (ABC) containing 10% acetonitrile. After three additional washes with 50 mM ABC/10% acetonitrile, which were performed by

centrifugation to ensure quantitative removal of liquids potentially remaining underneath the ultrafiltration membrane, proteins were digested overnight at 37°C with 400 ng trypsin in 40 µl of the same buffer. Tryptic peptides were recovered by centrifugation followed by two additional extraction steps with 40 µl of 50 mM ABC and 40 µl of 1% trifluoroacetic acid (TFA), respectively. Aliquots of the combined flow-throughs were spiked with 10 fmol/µl of yeast enolase-1 tryptic digest standard (Waters Corporation) for quantification purposes and directly subjected to analysis by liquid chromatography coupled to electrospray mass spectrometry (LC-MS). A pool of all samples was injected at least before and after any sample set to monitor stability of instrument performance.

## Mass spectrometry

Nanoscale reversed-phase UPLC separation of tryptic peptides was performed with a nanoAcquity UPLC system equipped with a Symmetry C18 5 µm, 180 µm × 20 mm trap column and a HSS T3 C18 1.8 µm, 75 µm × 250 mm analytical column (Waters Corporation) maintained at 45°C. Injected peptides were trapped for 4 min at a flow rate of 8 µl/min 0.1% TFA and then separated over 120 min at a flow rate of 300 nl/min with a gradient comprising two linear steps of 3–35% mobile phase B in 105 min and 35–60% mobile phase B in 15 min, respectively. Mobile phase A was water containing 0.1% formic acid while mobile phase B was acetonitrile containing 0.1% formic acid. Mass spectrometric analysis of tryptic peptides was performed using a Synapt G2-S quadrupole time-of-flight mass spectrometer equipped with ion mobility option (Waters Corporation). Positive ions in the mass range m/z 50 to 2000 were acquired with a typical resolution of at least 20.000 FWHM (full width at half maximum) and data were lock mass corrected post-acquisition. UDMS$^E$ and DRE-UDMS$^E$ analyses were performed in the ion mobility-enhanced data-independent acquisition mode with drift time-specific collision energies as described in detail by *Distler et al. (2016)* and *Distler et al. (2014b)*. Specifically, for DRE-UDMS$^E$ a deflection device (DRE lens) localized between the quadrupole and the ion mobility cell of the mass spectrometer was cycled between full (100% for 0.4 s) and reduced (5% for 0.4 s) ion transmission during one 0.8 s full scan. Continuum LC-MS data were processed for signal detection, peak picking, and isotope and charge state deconvolution using Waters ProteinLynx Global Server (PLGS) version 3.0.2 (47). For protein identification, a custom database was compiled by adding the sequence information for yeast enolase 1 and porcine trypsin to the UniProtKB/Swiss-Prot mouse proteome and by appending the reversed sequence of each entry to enable the determination of false discovery rate (FDR). Precursor and fragment ion mass tolerances were automatically determined by PLGS 3.0.2 and were typically below 5 ppm for precursor ions and below 10 ppm (root mean square) for fragment ions. Carbamidomethylation of cysteine was specified as fixed and oxidation of methionine as variable modification. One missed trypsin cleavage was allowed. Minimal ion matching requirements were two fragments per peptide, five fragments per protein, and one peptide per protein. The FDR for protein identification was set to 1% threshold.

## Analysis of proteomic data

For each genotype (*Prx$^{-/-}$* and wild type control mice sacrificed at P21), biochemical fractions enriched for PNS myelin were analyzed as three biological replicates (n = 3 per condition); each biological replicate representing a pool of 20 sciatic nerves dissected from 10 mice. The samples were processed with replicate digestion and injection, resulting in four technical replicates per biological replicate and thus a total of 12 LC-MS runs per condition to be compared, essentially as previously reported for CNS myelin (*Patzig et al., 2016b*; *Erwig et al., 2019b*). The freely available software ISOQuant (www.isoquant.net) was used for post-identification analysis including retention time alignment, exact mass and retention time (EMRT) and ion mobility clustering, peak intensity normalization, isoform/homology filtering and calculation of absolute in-sample amounts for each detected protein (*Distler et al., 2016*; *Distler et al., 2014b*; *Kuharev et al., 2015*) according to the TOP3 quantification approach (*Silva et al., 2006*; *Ahrné et al., 2013*). Only peptides with a minimum length of seven amino acids that were identified with scores above or equal to 5.5 in at least two runs were considered. FDR for both peptides and proteins was set to 1% threshold and only proteins reported by at least two peptides (one of which unique) were quantified using the TOP3 method. The parts per million (ppm) abundance values (i.e. the relative amount (w/w) of each protein in respect to the sum over all detected proteins) were log2-transformed and normalized by subtraction

of the median derived from all data points for the given protein. Significant changes in protein abundance were detected by moderated t-statistics essentially as described (*Ambrozkiewicz et al., 2018*; *Erwig et al. (2019b)* ) across all technical replicates using an empirical Bayes approach and false discovery (FDR)-based correction for multiple comparisons (*Kammers et al., 2015*). For this purpose, the Bioconductor R packages 'limma' (*Ritchie et al., 2015*) and 'q-value' (*Storey, 2003*) were used in RStudio, an integrated development environment for the open source programming language R. Proteins identified as contaminants (e.g. components of blood or hair cells) were removed from the analysis. Proteins with ppm values below 100 which were not identified in one genotype were considered as just above detection level and also removed from the analysis. The relative abundance of a protein in myelin was accepted as altered if both statistically significant (q-value <0.05). Pie charts, heatmaps and volcano plots were prepared in Microsoft Excel 2013 and GraphPad Prism 7. Pearson's correlation coefficients derived from log2-transformed ppm abundance values were clustered and visualized with the tool heatmap.2 contained in the R package gplots (CRAN.R-project.org/package = gplots). Only pairwise complete observations were considered to reduce the influence of missing values on clustering behavior. The mass spectrometry proteomics data have been deposited to the ProteomeXchange Consortium (proteomecentral.proteomexchange.org) via the PRIDE partner repository (*Vizcaíno et al., 2016*) with the dataset identifier PXD015960.

## Gel electrophoresis and silver staining of gels

Protein concentration was determined using the DC Protein Assay kit (BioRad). Samples were separated on a 12% SDS-PAGE for 1 hr at 200 V using the BioRad system, fixated overnight in 10% [v/v] acetic acid and 40% [v/v] ethanol and then washed in 30% ethanol (2 × 20 min) and ddH$_2$O (1 × 20 min). For sensitization, gels were incubated 1 min in 0.012% [v/v] Na$_2$S$_2$O$_3$ and subsequently washed with ddH$_2$O (3 × 20 s). For silver staining, gels were impregnated for 20 min in 0.2% [w/v] AgNO$_3$/ 0.04% formaldehyde, washed with ddH$_2$O (3 × 20 s) and developed in 3% [w/v] Na$_2$CO$_3$/0.02% [w/v] formaldehyde. The reaction was stopped by exchanging the solution with 5% [v/v] acetic acid.

## Immunoblotting

Immunoblotting was performed as described by *Schardt et al. (2009)* and *de Monasterio-Schrader et al. (2013)*. Primary antibodies were specific for dystrophin-related-protein 2 (DRP2; Sigma; 1:1000), peripheral myelin protein 2 (PMP2; ProteinTech Group 12717–1-AP; 1:1000), proteolipid protein (PLP/DM20; A431; *Jung et al., 1996*; 1:5000), Monocarboxylate transporter 1 (MCT1/ SLC16A1; *Stumpf et al., 2019*; 1:1000), periaxin (PRX; *Gillespie et al., 1994*; 1:1000), sodium/potassium-transporting ATPase subunit alpha-1 (ATP1A1; 1:2000; Abcam #13736–1-AP), myelin protein zero (MPZ/P0; *Archelos et al., 1993*; kind gift by J Archelos-Garcia; 1:10.000), voltage-dependent anion-selective channel protein (VDAC; Abcam #ab15895; 1:1000), basigin (BSG/CD147; ProteinTech Group #ab64616; 1:1000), neurofilament H (NEFH/NF-H; Covance #SMI-32P; 1:1000), voltage-gated potassium channel subunit A member 1 (KCNA1; Neuromab #73–007; 1:1000), EGR2/ KROX20 (*Darbas et al., 2004*; kind gift by D Meijer, Edinburgh; 1:1000) and myelin basic protein (MBP; 1:2000). To generate the latter antisera, rabbits were immunized (Pineda Antikörper Service, Berlin, Germany) with the KLH-coupled peptide CQDENPVVHFFK corresponding to amino acids 212–222 of mouse MBP isoform 1 (Swisprot/Uniprot-identifier P04370-1). Anti-MBP antisera were purified by affinity chromatography and extensively tested for specificity by immunoblot analysis of homogenate of brains dissected from wild-type mice compared to *Mbp*$^{shiverer/shiverer}$ mice that lack expression of MBP. Appropriate secondary anti-mouse or -rabbit antibodies conjugated to HRP were from dianova. Immunoblots were developed using the Enhanced Chemiluminescence Detection kit (Western Lightning Plus, Perkin Elmer) and detected with the Intas ChemoCam system (INTAS Science Imaging Instruments GmbH, Göttingen, Germany).

## Immunolabelling of teased fibers

Teased fibers were prepared as previously described by *Sherman et al. (2001)* and *Catenaccio and Court (2018)*. For each genotype, one male mouse was sacrificed by cervical dislocation at P17. Immunolabelling of teased fibers was performed as described by *Patzig et al. (2016b)*. Briefly, teased fibers were fixed for 5 min in 4% paraformaldehyde, permeabilized 5 min with ice-cold

methanol, washed in PBS (3 × 5 min) and blocked for 1 hr at 21°C in blocking buffer (10% horse serum, 0.25% Triton X-100, 1% bovine serum albumin in PBS). Primary antibodies were applied overnight at 4°C in incubation buffer (1.5% horse serum, 0.25% Triton X-100 in PBS). Samples were washed in PBS (3 × 5 min) and secondary antibodies were applied in incubation buffer (1 hr, RT). Samples were again washed in PBS (2 × 5 min), and 4',6-diamidino-2-phenylindole (DAPI; 1:50 000 in PBS) was applied for 10 min at RT. Samples were briefly washed 2x with ddH$_2$O and mounted using Aqua-Poly/Mount (Polysciences, Eppelheim, Germany). Antibodies were specific for myelin-associated glycoprotein (MAG clone 513; Chemicon MAB1567; 1:50) and MCT1/SLC16A1 (107). Secondary antibodies were donkey α-rabbit-Alexa488 (Invitrogen A21206; 1:1000) and donkey α-mouse-Alexa555 (Invitrogen A21202; 1:1000). Labeled teased fibers were imaged using the confocal microscope Leica SP5. The signal was collected with the objective HCX PL APO lambda blue 63.0. x1.20. DAPI staining was excited with 405 nm and collected between 417 nm - 480 nm. To excite the Alexa488 fluorophore an Argon laser with the excitation of 488 nm was used and the emission was set to 500 nm - 560 nm. Alexa555 was excited by using the DPSS561 laser at an excitation of 561 nm and the emission was set to 573 nm - 630 nm. To export and process the images LAS AF lite and Adobe Photoshop were used.

## mRNA abundance profiles

Raw data were previously established (*Fledrich et al., 2018*) from the sciatic nerves of wild type Sprague Dawley rats at the indicated ages (E21, P6, P18; n = 4 per time point). Briefly, sciatic nerves were dissected, the epineurium was removed, total RNA was extracted with the RNeasy Kit (Qiagen), concentration and quality (ratio of absorption at 260/280 nm) of RNA samples were determined using the NanoDrop spectrophotometer (ThermoScientific), integrity of the extracted RNA was determined with the Agilent 2100 Bioanalyser (Agilent Technologies) and RNA-Seq was performed using the Illumina HiSeq2000 platform. RNA-Seq raw data are available under the GEO accession number GSE115930 (*Fledrich et al., 2018*). For the present analysis, the fastqfiles were mapped to *rattus norvegicus* rn6 using Tophat Aligner and then quantified based on the Ensemble Transcripts release v96. The raw read counts were then normalized using the R package DESeq2. The normalized gene expression data was then standardized to a mean of zero and a standard deviation of one, therefore genes with similar changes in expression are close in the euclidian space. Clustering was performed on the standardized data using the R package mfuzz. Transcripts displaying abundance differences of less than 10% coefficient of variation were considered developmentally unchanged.

## Venn diagrams

Area-proportional Venn diagrams were prepared using BioVenn (*Hulsen et al., 2008*) at www.bio-venn.nl/.

## GO-term

For functional categorization of the myelin proteome the associated gene ontology terms were systematically analyzed on the mRNA abundance cluster using the Database for Annotation, Visualization and Integrated Discovery (DAVID; https://david.ncifcrf.gov). For comparison known myelin proteins according to literature were added.

## Histological analysis

*Prx*[-/-] and control mice were perfused at the indicated ages intravascularly with fixative solution (2.5% glutaraldehyde, 4% paraformaldehyde, 0.1 M sodium cacodylate buffer, pH 7.4). Quadriceps nerves were removed, fixed for 2 hr at room temperature, followed by 18 hr at 4°C in the same fixative, postfixed in OsO$_4$, dehydrated a graded series of ethanol, followed by propylene oxide and embedded in Araldite. All axons not associated with a Remak bundle were counted and categorized as myelinated or non-myelinated. All myelin profiles lacking a recognizable axon were counted. The total number of axons were counted on micrographs of toluidine blue stained Araldite sections (0.5 µm) of quadriceps nerves. Precise p-values for the quantitative comparison between Ctrl and *Prx*[-/-] mice were: Total number of axons (*Figure 6B*; Student's unpaired t-test): 2 mo p=0.01734; 4 mo p=2.1E-05; 9 mo p=0.007625; Number of myelinated axons (*Figure 6C*; Student's unpaired t-test): 2

mo p=0.00444; 4 mo p=2.12E-05; 9 mo p=0.005766; Number of empty myelin profiles (*Figure 6D*; Student's unpaired t-test): 2 mo p=0.004445; 4 mo p=0.001461; 9 mo p=0.000695; Axonal diameters (*Figure 6E–G*; two-sided Kolmogorow-Smirnow test): 2 mo p=2.20E-16; 4 mo p=2.20E-16; 9 mo p=2.20E-16.

## Acknowledgements

We thank J Archelos-Garcia and D Meijer for antibodies, T Buscham and J Edgar for discussions, L Piepkorn for support in data analysis, K-A Nave for support made possible by a European Research Council Advanced Grant ('MyeliNano' to K-AN) and the International Max Planck Research School for Genome Science (IMPRS-GS) for supporting SBS.

## Additional information

### Funding

| Funder | Grant reference number | Author |
| --- | --- | --- |
| Deutsche Forschungsgemeinschaft | WE 2720/2-2 | Hauke B. Werner |
| Deutsche Forschungsgemeinschaft | WE 2720/4-1 | Hauke B. Werner |
| Deutsche Forschungsgemeinschaft | WE 2720/5-1 | Hauke B. Werner |
| Deutsche Forschungsgemeinschaft | RO 4076/3-2 | Moritz J. Rossner |
| Wellcome | 0842424 | Peter Brophy |

The funders had no role in study design, data collection and interpretation, or the decision to submit the work for publication.

### Author contributions

Sophie B Siems, Performed all experiments not specified otherwise, conducted statistical analysis, contributed to analysis and interpretation of data; Olaf Jahn, Performed proteome analysis, contributed to analysis and interpretation of data and writing the article; Maria A Eichel, Performed teased fiber labeling and microscopy; Nirmal Kannaiyan, Performed bioinformatic analysis of RNA-Seq data; Lai Man N Wu, Diane L Sherman, Performed histological analysis; Kathrin Kusch, Provided unpublished reagents; Dörte Hesse, Contributed to proteome analysis; Ramona B Jung, Performed biochemical purification of myelin; Robert Fledrich, Michael W Sereda, Provided an unpublished RNA-Seq dataset; Moritz J Rossner, Supervised bioinformatic analysis of RNA-Seq data; Peter J Brophy, Supervised histological analysis; Hauke B Werner, Conceived, designed and directed the study, analyzed and interpreted data and wrote the article

### Author ORCIDs

Olaf Jahn (iD) https://orcid.org/0000-0002-3397-8924
Kathrin Kusch (iD) https://orcid.org/0000-0002-5079-502X
Hauke B Werner (iD) https://orcid.org/0000-0002-7710-5738

### Ethics

Animal experimentation: All animal work conformed to United Kingdom legislation (Scientific Procedures) Act 1986 and to the University of Edinburgh Ethical Review Committee policy; Home Office project license No. P0F4A25E9.

### Decision letter and Author response

Decision letter https://doi.org/10.7554/eLife.51406.sa1
Author response https://doi.org/10.7554/eLife.51406.sa2

## Additional files

### Supplementary files
• Transparent reporting form

### Data availability

All data generated or analysed during this study are included in the manuscript and supporting files. This includes the mass spectrometry proteomics data. Source data files have been provided for Figures 1, 3 and 5. Additional to being provided in the source data files, mass spectrometry proteomics data have been deposited to the PRIDE/ProteomeXchange Consortium with dataset identifier PXD015960.

The following dataset was generated:

| Author(s) | Year | Dataset title | Dataset URL | Database and Identifier |
|---|---|---|---|---|
| Olaf Jahn | 2020 | Proteome profile of peripheral myelin in healthy mice and in a neuropathy model | https://www.ebi.ac.uk/pride/archive/projects/PXD015960 | PRIDE, PXD015960 |

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
