## [Decision Letter]

**Acceptance summary:**

The authors used nano UPLC and quantitative mass spectrometry to measure the PNS myelin proteome from the sciatic nerve of healthy wild type mice and in periaxin (Prx)^-/-^ mice, an established peripheral neuropathy model of Charcot Marie Tooth disease. They also integrate and compare proteomic data with myelin transcriptome using published RNA sequencing data. This comprehensive proteomic and transcriptomic analyses of peripheral myelin will be an important resource for the field.

**Decision letter after peer review:**

Thank you for submitting your article "Proteome profile of peripheral myelin in healthy mice and in a neuropathy model" for consideration by *eLife*. Your article has been reviewed by three peer reviewers, one of whom is a member of our Board of Reviewing Editors, and the evaluation has been overseen by Gary Westbrook as the Senior Editor. The reviewers have opted to remain anonymous.

The reviewers have discussed the reviews with one another and the Reviewing Editor has drafted this decision to help you prepare a revised submission.

Summary:

This is an interesting Tools and Resources article that expands and refines earlier work, mainly using proteomics methods.Here the authors used nanoUPLC and quantitative mass spectrometry to measure the PNS myelin proteome from the sciatic nerve of healthy wild type mice and in Prx^-/-^ mice, an established peripheral neuropathy model of Charcot Marie Tooth disease. They also integrate and compare proteomic data with myelin transcriptome using published RNA sequencing data. The main progress to the earlier work lies in the development of an integrated pipeline/workflow to do routine comparative analyses of peripheral nerve myelin proteoms with state-of-the-art tools.

A comprehensive proteomics and transcriptomics analyses of peripheral myelin will be an important resource for the field. However, there are several unsupported claims about the mass spectrometry and proteomics data and analyses that need to be addressed as outlined below. More quantification, validation and quality control is needed, in addition to addressing the specific essential points below.

Essential revisions:

1) Data quality and reproducibility needs to be shown.

– Pearson correlation between replicates and between data acquisition methods needs to be shown.

– Criteria other than number of proteins identified should be rigorously evaluated and stated.

– Intensities across runs for spike-in protein, pre- and post-normalization should be shown.

2) Validation of key findings is needed.

– Looking at the Westerns there does seem to be a depletion of lysate proteins but by no means complete. What work was done to draw an enrichment threshold over a control enrichment like lysate to show the proteins identified were myelin specific?

– To claim that the abundance of MCT1/SLC16A1 is "strongly reduced" in prx^-/-^ mice, please quantify the Western blots, and perform statistics with the necessary biological replicates to do so for this and other differentially expressed proteins.

3) For data-independent acquisition (DIA) analysis, please show how the library was made and what its quality is.

4) References for UDMS, etc. need to be cited. These are not common nomenclature.

5) All of the data needs to be made available for the reviewers.

6) Figure 5D does not show a significance threshold.

[Editors' note: further revisions were suggested prior to acceptance, as described below.]

Thank you for resubmitting your work entitled "Proteome profile of peripheral myelin in healthy mice and in a neuropathy model" for further consideration by *eLife*. Your revised article has been evaluated by Gary Westbrook (Senior Editor) and a Reviewing Editor. The manuscript has been improved but there are remaining issues that we think need to be addressed before acceptance – to focus the study and to explore and show the quality of the quantitation.

Summary:

Although this study of myelin subproteomes is potentially a useful resource for the neuroscience community, we think the data acquisition methods are compared in a somewhat cursory fashion and several issues remain as summarized below.

– Clustering of published RNA data sets is used, though they are not compared to the proteomic data generated.

– The comparison between myelin proteomes between wt and Prx KOs is of biological interest, but the comparison is not done in manners consistent within the study or with current quantitative proteomic studies.

Major comments:

1) The comparison between data acquisition methods needs to be either better justified. If DRE-UDMS is quantitative (which there is no strong data in this study to support this) and IDs the most proteins, please report that. There is no reason to show the other two. In fact, according to Figure 1C MBP has almost a two order of magnitude difference between MSE and UDMS. With spike-ins, if used appropriately, this should not be the case. So why include them? The authors refer to one "correct quantitation", one "deep quantitation" and one for "differential analysis". If the data were in fact quantitative you should be able to show the data is accurate and precise, and then do differential analysis with the same data.

2).All protein abundance metrics should be relative to an enriched proteins. If one does an upfront enrichment, the efficiency could differ so the absolute mols is not very meaningful. But if it was all relative to some myelin protein, the relative values would be more intuitive. Figure 2 is the preferable way to show this.

3) Why cluster the RNA data vs compare those clusters to proteomic data, or integrate the data sets in some way? It is not clear that we are learning anything new from the proteomic analysis as presented. Please address.

It is peculiar that Prx was not differentially abundant their heat map or the volcano plots. There are have several cases where proteins like DRP2 or PYGM were not detected in all replicates. Other proteins seem to lose all signal (AP2A1, ATP1A4 and PLCD1). Please confirm and address.

4) Along these lines, please revise Figure 5C to be the top 40 proteins statistically different between the samples. vs selecting top/bottom 40 proteins according to fold change with the quant is variable at best (VCL, NEFL, APOA1 as examples) or those not detected in more than one replicate.

5) Figure 5A is described to not have any major differences other than PRX's absence. Yet 5D seems to show most proteins are differentially expressed if the dashed line is your statistical cutoff. Please address.

---

## [Author Response]

Essential revisions:1) Data quality and reproducibility needs to be shown.– Pearson correlation between replicates and between data acquisition methods needs to be shown.

In response, we have now included all Pearson’s correlation coefficients as a clustered heatmap in the new Figure 1—figure supplement 1. The respective methodology has been included in the Materials and methods section.

– Criteria other than number of proteins identified should be rigorously evaluated and stated.

In response, we have now included average sequence coverage in the main text where numbers of quantified proteins are given, as this is probably the most relevant criterion for a biology-oriented readership. Further details including the number of identified peptides are now included as additional sheets in Figure 1—source data 1, Figure 5—source data 1 and Figure 5—source data 2.

– Intensities across runs for spike-in protein, pre- and post-normalization should be shown.

This request, together with comments 3 and 4 below, led us to conclude that in the original manuscript we may have failed to absolutely clearly convey some of the key features of our proteomic workflow. Briefly, these are:

i) data-independent acquisition of all precursors and their fragments with a wide band-pass filter for precursor selection in an alternating low and elevated energy mode across the full mass range (referred to in the field as MS^E^),

ii) analysis of these complex data by dedicated search algorithms that do not require spectral libraries but instead make use of retention and drift time profiles (when ion mobility separation of precursors is applied) for deconvolution of the precursor-fragment ion alignment, and

iii) inference of protein abundance values by reference to a spiked-in tryptic digest of a standard protein (yeast enolase) of known amount on the basis of the proven correlation between average intensity of three most intense peptides and the absolute amount of their source protein, referred to in the field as TOP3 (or Hi3) quantification (also see next paragraphs).

In response, aiming to explain these features better, we modified our Introduction, i.e. extended our explanation and included additional references including review articles. We hope that we thereby provide better coverage of our data acquisition and analysis (above points i and ii). To address potential confusion regarding protein quantification (above point iii), we have now explained and referenced the TOP3 method in the second paragraph of the Results section.

As to the intensities of the spike protein, we are not certain if this request arose from shortcomings in our initial description of our workflow or if it is rather related to the TOP3 approach. Indeed, we spiked a tryptic digest (yeast enolase 1 at 10 fmol/ul) into the actual sample digest so that the amount of all proteins identified can be obtained by reference to the fixed on-column fmol amount of the spike.

In response, we have added pre- and post-normalization TOP3 values for yeast enolase 1 to the protein identification details (sheet 1 of the respective source data table, see above). Moreover, we have reintroduced the ppm abundance values of yeast enolase 1 and porcine trypsin into Figure 1—source data 1, Figure 5—source data 1 and Figure 5—source data 2, from which they were previously filtered as exogenously added proteins. If the intention of the reviewer was to use the spike values for judging on reproducibility between runs, we note that this aspect in now also addressed by the Pearson’s correlation heatmap in the new Figure 1—figure supplement 1.

2) Validation of key findings is needed.– Looking at the Westerns there does seem to be a depletion of lysate proteins but by no means complete. What work was done to draw an enrichment threshold over a control enrichment like lysate to show the proteins identified were myelin specific?

Indeed, biochemical fractions purified from biological samples are never absolutely pure, and myelin is no exception. We consider a threshold factor of ≥ 1.5-fold as enrichment and a factor of ≤ 0.75-fold as depletion. Yet, to claim that a protein is “myelin-specific” additional to biochemical methods, in our view requires more direct visualization of its localization, e.g. by immuno-histochemistry. Accordingly, selected references for “proven myelin proteins” are given in Table 1. We also provide heatmaps of the quantifications of the immunoblots in Figure 1B and Figure 5E in Author response image 1. However, upon careful consideration we chose to not include these heatmaps into our manuscript considering that we believe that they do not provide additional relevant information. However, we will include the heatmaps into our manuscript if requested by the Editors.

– To claim that the abundance of MCT1/SLC16A1 is "strongly reduced" in prx^-/-^ mice, please quantify the Western blots, and perform statistics with the necessary biological replicates to do so for this and other differentially expressed proteins.

In response, when describing the MCT1 immunoblot in the main text we changed the term “validate the proteome results” to “confirm the proteome results”. We also provide heatmaps of the quantifications of the immunoblots in Figure 5E in Author response image 1. However, upon careful reflection we chose to not include these heatmaps into our manuscript considering that we believe that they do not provide additional relevant information. However, we will include the heatmaps into our manuscript if requested by the editors.

**Author response image 1. respfig1:** Quantification of immunoblots in Figure 1B and Figure 5E.

For statistical assessment of the differential proteome analysis, kindly see Figure 5C showing three biological replicates in a heatmap as well as Figure 5—source data 2 giving the PPM values for all replicates and all q-values. Both representations show the reduction of the abundance of MCT1 (and other proteins). Yet, we would like to argue that for quantitative assessment, proteome analysis using MS^E^-based methods provide superior precision compared to immunoblotting. Indeed, we perform immunoblotting (as in Figure 5E) for principal confirmation of findings rather than a second method toward quantification. The immunoblot in Figure 5E has been performed with two biological replicates, which we believe is sufficient for confirmation of a principal finding even if it does not allow statistical assessment. For example, the immunoblots clearly confirm the reduction of the abundance of MCT1 in *Prx*-mutant myelin; the result is additionally confirmed by immunohistochemistry (Figure 5F). Considering that the relevant myelin samples have been used up by now, it would require breeding additional 30 mice per genotype for repeating these immunoblots with three biological replicates. We would like to ask for the reviewer‘s understanding that we believe this somewhat unproportional, in particular when considering that Prx-mutant mice as a neuropathy model are burdened. Indeed, we find it unlikely that the local animal ethics authorities would approve a request for an animal experiment license for this purpose. Together, we would like to ask for the reviewer’s understanding that we did not repeat the immunoblots in Figure 5E.

3) For data-independent acquisition (DIA) analysis, please show how the library was made and what its quality is.

In response, we now describe the DIA approach in more detail (see our response to comment 1, bullet point 3). We note that searching against spectral libraries is not a feature, different from SWATH approaches. Instead, in the MS^E^ approach, the highly complex fragmentation spectra are deconvoluted by chromatographic profiles. In the UDMS^E^ approach, separation of precursors by ion mobility is used in addition. In both cases, proteins are identified by using dedicated search algorithms to directly search against FASTA database files. We hope that our modifications of the main text and the inclusion of additional references address any potential sources of confusion.

4) References for UDMS, etc. need to be cited. These are not common nomenclature.

In response, we have now included additional references where appropriate. We have also improved the explanations of the abbreviations and basic principles of the three utilized data acquisition modes where appropriate.

5) All of the data needs to be made available for the reviewers.

All datasets were submitted to the PRIDE/ProteomeXchange Consortium with dataset identifier PXD015960 and are available to the reviewers.

6) Figure 5D does not show a significance threshold.

We thank the reviewers for the observation that the significance threshold (although shown in the figure) was previously not explained in the figure legend. In response, thus, we have now included in the figure legend the statement “Stippled lines mark a -log10-transformed q-value of 1.301, reflecting a q-value of 0.05 as significance threshold”.

[Editors' note: further revisions were suggested prior to acceptance, as described below.]

Major comments:1) The comparison between data acquisition methods needs to be either better justified. If DRE-UDMS is quantitative (which there is no strong data in this study to support this) and IDs the most proteins, please report that. There is no reason to show the other two. In fact, according to Figure 1C MBP has almost a two order of magnitude difference between MSE and UDMS. With spike-ins, if used appropriately, this should not be the case. So why include them? The authors refer to one "correct quantitation", one "deep quantitation" and one for "differential analysis". If the data were in fact quantitative you should be able to show the data is accurate and precise, and then do differential analysis with the same data.

In response we have completely re-written the relevant part of the Results section (“Proteome analysis of peripheral myelin”),aiming at a better conveyance of the relevance of each of the three data acquisition modes in the context of the specific challenge that the myelin proteome is dominated by exceptionally abundant myelin proteins. In particular when considering that this is a Tools and Resources manuscript, we are convinced that comparing different data acquisition modes for their advantages and limitations, and developing a novel one for routine myelin proteome profiling is a strength of our study, as we hope becomes obvious from the re-worked text.

2) All protein abundance metrics should be relative to an enriched proteins. If one does an upfront enrichment, the efficiency could differ so the absolute mols is not very meaningful. But if it was all relative to some myelin protein, the relative values would be more intuitive. Figure 2 is the preferable way to show this.

In response we have redone Figure 1C. Indeed, we agree that showing the log10 transformed data in amol on the Y-axis in the previous plot was somewhat unfortunate as these values may vary if protein load differs between conditions. The Y-axis now displays parts per million (ppm) abundance values, i.e. the relative amount (w/w) of each protein in respect to the sum over all proteins detected in the actual data acquisition mode. We feel that showing the data relative to total myelin protein is more reliable than relative to a single myelin protein, the amount of which may be prone to technical variations as well.

3) Why cluster the RNA data vs compare those clusters to proteomic data, or integrate the data sets in some way? It is not clear that we are learning anything new from the proteomic analysis as presented. Please address.

In response we have re-written the relevant part of Results section and figure legend, aiming to better convey our strategy why and how to present the mRNA data. Indeed, we have used our proteomic inventory of the proteins associated with peripheral myelin to filter the mRNA data from peripheral nerves. Rather than displaying all >10.000 transcripts expressed in the peripheral nerve we here display selectively only those about 1.000 transcripts of which the protein product was identified in peripheral myelin. By this strategy we discriminate myelin-related mRNAs from other mRNAs in peripheral nerves, such as those present in axons, fibroblasts, immune cells, blood cells, endothelium, etc.. Although we agree that our filtering strategy may not necessarily be viewed as “comparison” or “integration” in the conventional sense, we hope to convince the reviewer that this conveys relevant information about the developmental abundance profiles of the transcripts encoding myelin-associated proteins.

It is peculiar that Prx was not differentially abundant their heat map or the volcano plots. There are have several cases where proteins like DRP2 or PYGM were not detected in all replicates. Other proteins seem to lose all signal (AP2A1, ATP1A4 and PLCD1). Please confirm and address.4) Along these lines, please revise Figure 5C to be the top 40 proteins statistically different between the samples. vs selecting top/bottom 40 proteins according to fold change with the quant is variable at best (VCL, NEFL, APOA1 as examples) or those not detected in more than one replicate.

In response we have redone Figure 5C and include the new Figure 5—figure supplement 1B. The heatmap in Figure 5C is now sorted by q-value (instead of before by fold change) and displays the fold-change of all 12 technical replicates (instead of before the average fold change of the three biological replicates). The volcano plot in Figure 5—figure supplement 1B uses axis scales different from those in Figure 5D to allow representing the trace amount of PRX peptides detected in mutant myelin. Both representations include AP2A1, ATP1A4 and PLCD, which indeed were not identified in mutant myelin. We note however that the abundance of these proteins in WT myelin is relatively low, different from the more abundant DRP2.

5) Figure 5A is described to not have any major differences other than PRX's absence. Yet 5D seems to show most proteins are differentially expressed if the dashed line is your statistical cutoff. Please address.

In response, we have re-phrased the part of the Results section that describes the silver gel (Figure 5A). Indeed, numerous bands display genotype-dependent intensity differences. However, except for the most abundant proteins (MPZ/P0, MBP, PRX), for most bands it is not possible to say which proteins are the major constituents at the level of the silver gel, necessitating differential proteome analysis.